# Hippocampal GABA enables inhibitory control over unwanted thoughts

Taylor W. Schmitz[1,2], Marta M. Correia[1,3], Catarina S. Ferreira[4], Andrew P. Prescot[5] & Michael C. Anderson[1,6]

Intrusive memories, images, and hallucinations are hallmark symptoms of psychiatric disorders. Although often attributed to deficient inhibitory control by the prefrontal cortex, difficulty in controlling intrusive thoughts is also associated with hippocampal hyperactivity, arising from dysfunctional GABAergic interneurons. How hippocampal GABA contributes to stopping unwanted thoughts is unknown. Here we show that GABAergic inhibition of hippocampal retrieval activity forms a key link in a fronto-hippocampal inhibitory control pathway underlying thought suppression. Subjects viewed reminders of unwanted thoughts and tried to suppress retrieval while being scanned with functional magnetic resonance imaging. Suppression reduced hippocampal activity and memory for suppressed content. [1]H magnetic resonance spectroscopy revealed that greater resting concentrations of hippocampal GABA predicted better mnemonic control. Higher hippocampal, but not prefrontal GABA, predicted stronger fronto-hippocampal coupling during suppression, suggesting that interneurons local to the hippocampus implement control over intrusive thoughts. Stopping actions did not engage this pathway. These findings specify a multi-level mechanistic model of how the content of awareness is voluntarily controlled.

[1] MRC Cognition and Brain Sciences Unit, Cambridge CB2 7EF, UK. [2] Wolfson College, University of Cambridge, Cambridge CB3 9BB, UK. [3] Downing College, University of Cambridge, Cambridge CB2 1DQ, UK. [4] Research Center for Mind, Brain and Behavior, University of Granada, Granada 18011, Spain. [5] School of Medicine, University of Utah, Salt Lake City, Utah 84132, USA. [6] Behavioural and Clinical Neurosciences Institute, University of Cambridge, Cambridge CB2 3EB, UK. Correspondence and requests for materials should be addressed to T.W.S. (email: tws35@cam.ac.uk) or to M.C.A. (email: Michael.Anderson@mrc-cbu.cam.ac.uk)

Intrusive memories, hallucinations, ruminations, and persistent worries lie at the core of conditions such as post-traumatic stress disorder, schizophrenia, major depression, and anxiety[1–4]. These debilitating symptoms are widely believed to reflect, in part, the diminished engagement of the lateral prefrontal cortex to stop unwanted mental processes, a process known as inhibitory control[5–12]. However, these disorders share another feature of their pathophysiology that is not usually considered theoretically relevant to control: hippocampal hyperactivity[13–20]. In this article, we examine why this recurring feature, rarely considered by researchers interested in cognitive control, is often strongly related to the occurrence and frequency of intrusive symptomatology. In so doing, we provide evidence for a mechanism enabling inhibitory control over thought: GABAergic inhibition of hippocampal activity.

In individuals with schizophrenia, the severity of positive symptoms, such as hallucination, increases with hippocampal hyperactivity, as indexed from abnormally elevated resting blood oxygen-level-dependent (BOLD) activity, or increased regional cerebral blood flow, blood volume, or blood glucose metabolic rate[14, 19]. Evidence indicates that such hyperactivity arises in part from dysfunctional GABAergic interneurons, and post-mortem anatomical studies confirm substantial hippocampal parvalbumin-positive and somatostatin-positive interneuron loss in victims of the disease[13–15]. Consistent with this view, animal models of schizophrenia show that disrupting GABAergic inhibition in the hippocampus by transgenic or pharmacological manipulations reliably reproduces hippocampal hyperactivity and volume loss, along with behavioral phenomena paralleling symptoms present in this disorder[21, 22]. Interestingly, abnormally elevated hippocampal activity also occurs in post-traumatic stress disorder and major depression, and this pattern predicts both flashback intensity and depressive rumination[10, 16–18, 20]. In both of these disorders, impaired GABAergic inhibition in the hippocampus could contribute to these symptoms, possibly by a cascade of processes initiated by stress[23]. Indeed, animal models of anxiety often focus on compromised GABAergic inhibition in the hippocampus, which produces symptoms consistent with a dysregulation in affective control, including impaired extinction of conditioned fear[24–27]. Together, these findings suggest that a deficit of GABAergic inhibition local to the hippocampus contributes to problems controlling a spectrum of intrusive memories and thoughts, although the pathogenesis of this deficit and its specific manifestations across disorders may vary. The basic link between hippocampal GABA and the capacity to control unwanted thoughts, however, remains unexplored.

Here we test a novel hypothesis about how hippocampal GABA supports this core feature of voluntary control over the contents of awareness. We hypothesized that GABAergic inhibition in the hippocampus forms a critical link in a fronto-hippocampal inhibitory control pathway that suppresses unwanted thoughts. Observations from both human neuroimaging and rodent electrophysiology motivate this hypothesis. Human imaging studies indicate that when individuals are given a reminder to an unwanted thought and try to suppress the thought from awareness, the right dorsolateral prefrontal cortex (DLPFC) acts, via polysynaptic pathways[28], to downregulate hippocampal activity, inducing forgetting of suppressed content[29–35]. This provides a systems level model for how thought suppression occurs. Rodent electrophysiology, on the other hand, demonstrates that tonically disinhibiting GABAergic interneuron networks in the hippocampus desynchronizes hippocampal rhythms, reducing overall activity and impairing memory function[36, 37]. Taken together, these observations raise the possibility that suppressing retrieval to stop an unwanted thought recruits a fronto-hippocampal inhibitory control pathway that engages this

hippocampal GABAergic mechanism. Specifically, prefrontal control signals may tonically increase activity in local hippocampal interneuron networks, inhibiting (and desynchronizing) principal cell activity throughout the hippocampus, impairing retrieval and disrupting memory. If this hypothesis is correct, diminished GABAergic tone local to the hippocampus may mute the inhibitory impact of control signals originating from DLPFC, compromising the ability to suppress unwanted content. This same deficit of GABAergic tone may also cause abnormally elevated hippocampal activity (hippocampal hyperactivity), explaining the recurring association between this feature and intrusive symptomatology.

To address this hypothesis, we combined an established cognitive manipulation for measuring the ability to suppress unwanted thoughts, the Think/No-Think paradigm[28–35, 38], with both functional magnetic resonance imaging (fMRI) and $^1$H magnetic resonance spectroscopy (MRS). This multimodal neuroimaging strategy provided, within the same individuals, co-localized in vivo measures of hippocampal BOLD response and GABA concentration. To address whether hippocampal BOLD–GABA relationships are anatomically specific, we also measured GABA and BOLD in the right DLPFC region thought to drive top-down control in the putative fronto-hippocampal pathway, and in a visual cortical control region outside of this pathway. To establish whether hippocampal GABA plays a functionally specific role in inhibiting thoughts, participants also performed a motor action inhibition task[39, 40] while fMRI was acquired, providing an index of inhibitory control over actions rather than thoughts. We, therefore, sought to determine whether hippocampal GABA selectively enables the control of unwanted thoughts, and if this arises because hippocampal GABA alters the impact of the putative fronto-hippocampal inhibitory control pathway. We found that higher GABA concentrations local to the hippocampus predicted superior forgetting of the thoughts that people tried to suppress, and, critically, the ability of the prefrontal cortex to exert long-range control over hippocampal retrieval processes. In contrast, hippocampal GABA predicted neither stopping ability nor hippocampal BOLD responses when people exerted inhibitory control over action. Our findings are consistent with the possibility that hippocampal GABA may play an important role in enabling the prefrontal cortex to suppress unwanted thoughts.

## Results

**Thought suppression engages a functionally specific pathway.**
Twenty-four healthy young adults performed adapted versions of the Think/No-Think (TNT)[38] and stop signal (SS)[39, 40] tasks, which were interleaved in a mixed block/event-related design (see Methods section). We focus first on the TNT task used to measure thought suppression. Prior to scanning, participants were drilled on a large set of word pairs, each one composed of a reminder and its associated thought. During scanning, on each trial, participants viewed one of these reminders, by itself. For each reminder, we cued participants either to retrieve its associated thought (Think trials), or instead to suppress its retrieval, stopping the thought from coming to mind at all (No-Think trials).

Previous work with the TNT paradigm establishes that suppressing retrieval of an associated thought downregulates hippocampal activity and impairs later memory for the suppressed content[28–35, 38, 41–44]. These hemodynamic and behavioral effects occur with a broad range of stimuli, including neutral or unpleasant words[29–31, 38, 42], visual objects[34], neutral or unpleasant scenes[32, 33, 41, 43, 44], autobiographical memories[45], and person-specific fears about their future[35]. Critically, populations that suffer from persistent intrusive thoughts such as those with post-traumatic stress disorder (PTSD)[44], depression[46, 47],

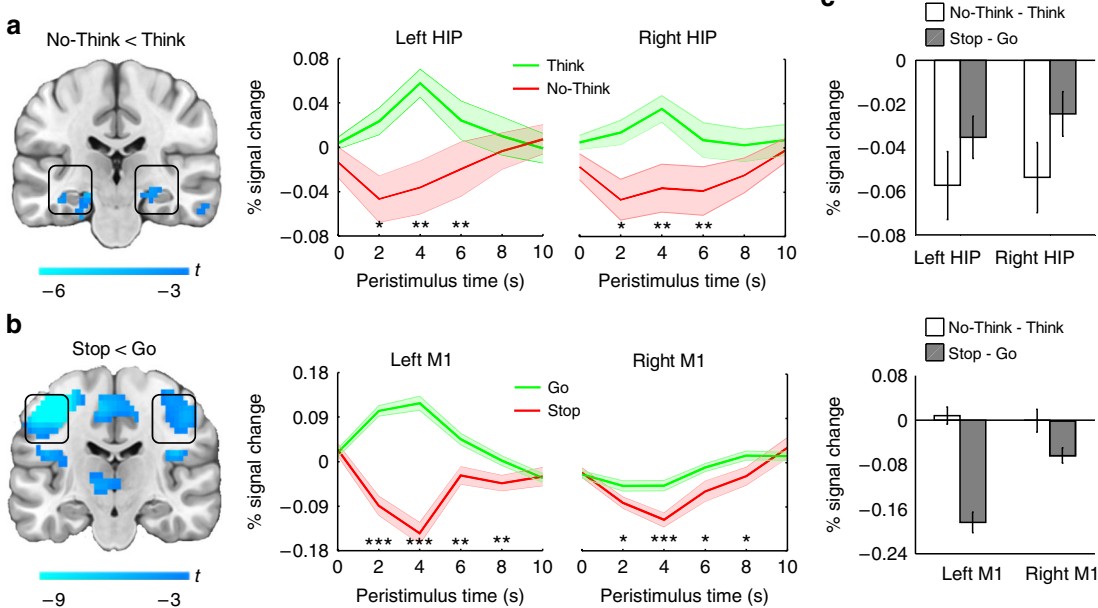

**Fig. 1** Domain-specific modulation during thought and action suppression. **a** and **b**. Group ($N = 24$) whole-brain contrasts for No-Think < Think (top) and Stop < Go (bottom). Thought suppression modulated bilateral hippocampal (HIP) activity. Action-stopping-modulated activity in primary motor cortex (M1), lateralized to the left (contralateral to hand) hemisphere. Boxes illustrate HIP and M1 activations on a coronal slice in MNI space. Activations are derived from an uncorrected cluster-defining threshold ($p < 0.001$), with cluster level false discovery rate $p < 0.05$. Color bars demarcate $T$-statistics. (Middle panels) A priori region of interest (ROI) analyses: Group hemodynamic time-courses were attenuated in HIP by thought suppression (No-Think) and in M1 by action suppression (Stop) relative to Think and Go, respectively ***$p < 0.001$; **$p < 0.01$; *$p < 0.05$. **c** Modality-dependent hemodynamic attenuation in HIP (top) and M1 (bottom) was confirmed with a repeated measures ANOVA, which revealed an ROI by Modality interaction. Error bars represent SEM

and anxiety[48] show significant deficits in suppression-induced forgetting. Individual differences in suppression-induced forgetting have been found to predict the frequency of naturally occurring traumatic intrusions in healthy individuals[49] and in PTSD[44], scores on clinical scales of ruminative thinking[50] and anxiety[35, 48], and measures of the general ability to control intrusive thoughts in daily life[51]. Together, these observations point to a general retrieval suppression mechanism that contributes to suppressing intrusive thoughts and suggest that these behavioral and hemodynamic effects index the efficiency of this mechanism.

To confirm these effects with the present stimuli, we compared BOLD responses between No-Think and Think trials in the anatomically defined right hippocampus region of interest (ROI)[52], and found that performing No-Think trials significantly reduced activation in this region ($t_{23} = 3.34$, $p = 0.003$; Fig. 1a), and found that performing No-Think trials significantly reduced effect was observed in the left hippocampus ($t_{23} = 3.69$, $p = 0.001$; Fig. 1a), though we focus on the right hippocampus ROI co-localized to our $^1H$ MRS acquisition. Suppressing retrieval also impaired participants' later memory for the suppressed items, demonstrating suppression-induced forgetting in this sample. Specifically, on a post-scan recall test, participants recalled No-think items less often (mean ± SEM: $59 \pm 3\%$) than they recalled either Think items ($65 \pm 3\%$; $t_{23} = 2.2$, $p = 0.04$) or Baseline items that they also learned, but that did not appear during the Think/No-Think phase (M = $65 \pm 3\%$; $t_{23} = 2.5$, $p = 0.02$). As previously shown[42], the amount of suppression-induced forgetting significantly increased with larger BOLD reductions during No-Think trials, though only in posterior hippocampus (Robust correlation[53]: $r = -0.56$, $t = -3.14$, 95% boot-strapped confidence interval (CI) (−0.84 to −0.14)). Together, these neural and behavioral markers of how well people suppressed unwanted thoughts confirm prior evidence for the role of the hypothesized fronto-hippocampal inhibitory control pathway in this function[29].

Alternating with blocks of the TNT task, participants also performed the SS task, a well-established procedure for measuring the inhibition of motor actions[39, 40] (see Methods section). We included the action-stopping task to contrast the effects of thought suppression with those of another widely studied inhibitory control task that should not rely on modulating hippocampal activity, but rather motor cortical activity. Prior to scanning, participants learned to press one of two buttons with their right index finger in response to differently colored circles. During scanning, participants performed a speeded motor response task that, on a minority of trials, required them to stop their motor action midstream if they received a stop signal. The right DLPFC (approximately Brodmann area 46) is thought to be critical for inhibitory control in a variety of cognitive task contexts[54]. To test whether this was indeed the case in our within-subjects study, we used an a priori ROI of the DLPFC (defined from a prior TNT study[31]) to extract BOLD response estimates during No-Think, Think, Stop, and Go trials. Consistent with a broad involvement in inhibitory control, DLPFC was significantly more engaged when either thoughts or actions needed to be inhibited (No-Think > Think, $t_{23} = 2.38$, $p = 0.026$; Stop > Go, $t_{23} = 4.32$, $p < 0.001$). To confirm that action-stopping targeted motor processes, we examined BOLD response in the hand lobule of left primary motor cortex (M1; defined with an independent localizer task; see Supplementary Methods). As predicted, when participants stopped a (right-handed) key press, we observed a significant downregulation of BOLD response in left M1 ($t_{23} = 10.02$, $p < 0.001$; Fig. 1b), consistent with prior findings[55]. The stopping-induced reduction in BOLD response (Stop < Go) was significantly larger in the left than in the right hemisphere ($t_{23} = 2.38$, $p = 0.026$), as would be expected, based on a right-handed key press response.

Critically, action stopping and thought suppression preferentially modulated the left M1 and hippocampus, respectively (Fig. 1c). In a Region (M1 vs. Hippocampus) by Modality

(Thoughts vs. Actions) by Task (Inhibition vs. Non-Inhibition) analysis of variance (ANOVA), there was a significant three way interaction in both the left and right hemispheres (Left: $F_{1,23} = 78.29$, $p < 0.001$; Right: $F_{1,23} = 13.56$, $p = 0.001$). Suppressing retrieval (No-Think < Think) evoked larger negative BOLD responses in the hippocampus compared to M1 (Left: $t_{23} = 5.80$, $p < 0.001$; Right: $t_{23} = 3.29$, $p = 0.003$). By contrast, suppressing motor actions (Stop < Go) evoked larger negative BOLD responses in M1 relative to the hippocampus (Left: $t_{23} = 5.80$, $p < 0.001$; Right: $t_{23} = 3.29$, $p = 0.003$). These differing modulatory profiles support the possibility that stopping thoughts engages a distinct fronto-hippocampal pathway that is not engaged by stopping actions. If so, GABA concentrations local to the hippocampus may be selectively tied to stopping thoughts, and not to stopping processes in general.

**Hippocampal GABA predicts successful thought suppression.** We next tested whether our hemodynamic and behavioral measures of thought suppression were related to hippocampal GABA. To do so, we employed [1]H MRS, a non-invasive imaging technique that provides sensitive measures of brain metabolites, such as GABA, by detecting the unique radio frequency signals arising from the hydrogen nuclear spins within these metabolites[56] (Methods section). In a separate MRS session, we quantified resting GABA concentrations in the hippocampus, the proposed site of inhibition, and in the right DLPFC, the proposed source of the control signal driving inhibitory activity in the hippocampus (see Fig. 2). As a control, we also measured GABA in the primary visual cortex, a region outside the proposed pathway (Supplementary Fig. 1). We used pre-defined anatomical landmarks to position the MRS ROIs for the DLPFC[31], hippocampus[52], and primary visual cortex[57], ensuring anatomical co-localization across subjects.

After applying MRS quality control standards to the data (see Methods), the final sample sizes for the [1]H MRS data were: Hippocampus ($n = 18$), DLPFC ($n = 23$), and visual cortex ($n = 20$). Mean GABA/Cre values ($\pm$SD) for the three MRS voxels were as follows: Hippocampus ($0.185 \pm 0.05$), DLPFC ($0.169 \pm 0.02$) and visual cortex ($0.192 \pm 0.05$). GABA concentrations were not correlated across our ROIs, as determined by robust correlation analyses[53]: Hippocampus and DLPFC ($r = -0.21$, $t < 1$); Hippocampus and Visual Cortex, ($r = -0.13$, $t < 1$); DLPFC and Visual Cortex, ($r = 0.04$, $t < 1$). The mean GABA/Cre value of 0.18 across these ROIS matches the reported value from an independent study using the same 2D [1]H MRS protocol ($0.18$)[57], as well as reported values from three other studies using similar protocols (mean GABA/Cre across ROIs and studies: $0.18$)[58–60]. These results suggest a level of reliability in GABA estimation close to that achieved by more frequently used [1]H MRS acquisition protocols, such as MEGA-PRESS[56]. Mean glutamate/Cre values ($\pm$SD) for the three ROIs were as follows: Hippocampus ($0.80 \pm 0.17$), DLPFC ($1.19 \pm 0.18$), and visual cortex ($1.07 \pm 0.11$). Mean gray matter concentration values ($\pm$SD) for the three ROIs were: Hippocampus ($65.4 \pm 5.29$), DLPFC ($28.4 \pm 4.02$), and visual cortex ($48.9 \pm 5.10$).

To examine relationships of [1]H MRS GABA with BOLD signal, and with behavior, we conducted a two-step procedure integrating robust correlation with partial correlation analyses. In the first step, we used a skipped correlation to derive a Pearson's $r$-value on data with bivariate outliers removed[53]. Outliers were determined automatically via an algorithm that found the central point in the distribution of data using the mid-covariance determinant. Orthogonal distances were then computed to this point, and any data outside the bound defined by the ideal estimator of the interquartile range was removed[53]. In the second step, we used

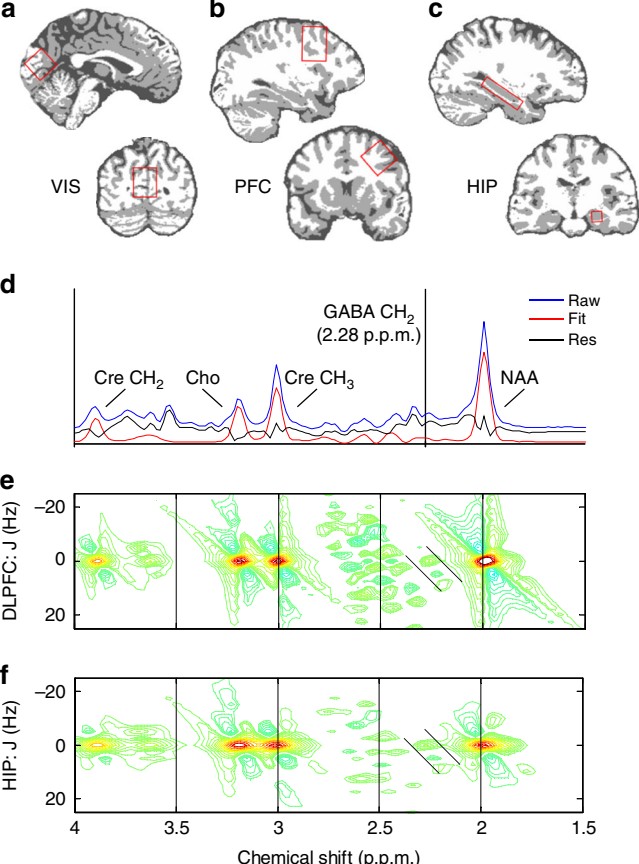

**Fig. 2** [1]H MRS quantification of GABA concentrations. **a–c** Positions of the visual cortical, DLPFC, and hippocampus (HIP) voxels are displayed on sagittal (top row) and coronal (bottom row) slices extracted from an example subject's tissue segmented structural scan. **d** An example of the [1]H MRS spectra displayed in one dimension. Blue line: raw metabolite spectra for an example subject. Red line: ProFit basis functions for singlet (one-peak) metabolites, including Creatine (Cre), Choline (Cho), and N-acetyl aspartate (NAA). Black line: residuals after fitting. Note the GABA CH2 methylene group at 2.28 PPM is invisible on the 1D plot. **e** and **f** Plotted for the DLPFC ($N = 23$) and HIP ($N = 18$) voxels are the fitted spectra (averaged overall subjects) of the same four metabolites, but now spread along two dimensions, the J-resolved axis ($\pm$20 Hz) plotted and the chemical shift axis (1.5–4 parts per million; p.p.m.). Both plots use identical scaling. Colors indicate minimum (blue) and maximum (red) height of spectral contours (arbitrary units). The GABA CH2 methylene group is visible at 2.28 p.p.m. (diagonal lines)

partial correlation to determine if any relationships observed in the robust estimation step were explained (or masked) by participant sex, the amount of gray matter volume captured by the [1]H MRS voxel, or co-localized concentrations of glutamate. We controlled for participant sex and gray matter tissue content in each ROI because these variables can influence estimates of GABA concentration[61]. Glutamate concentration was controlled because of the relationship of glutamatergic principal cell metabolism with BOLD[62] and GABA[63]. In both steps, inference of statistical significance was determined from 95% boot-strapped confidence intervals. These relationships are reported in Tables 1 and 2, and are described in the sections below.

Our main interest concerned whether task-induced changes in BOLD responses in the hippocampus were related to hippocampal GABA. Given the established role of the hippocampus in memory, mnemonic processes should drive changes in its activity, which in turn depends on the local population of GABAergic

**Table 1 Intermodal relationships of hippocampal GABA**

| Intermodal relationship | N | Robust | Out | Control | Functional specificity | Anatomical specificity |
|---|---|---|---|---|---|---|
| *A. HIP GABA: HIP BOLD* | | | | | | |
| No-Think | 18 | **−0.49*** | 2 | **−0.58*** | **−0.48*** | **−0.53*** |
| Think | 18 | **−0.47*** | 1 | **−0.48*** | **−0.61*** | **−0.59*** |
| Stop | 18 | −0.19 | 0 | −0.08 | – | – |
| Go | 18 | 0.23 | 0 | 0.15 | – | – |
| | | | | | | |
| *B. HIP GABA: Behavior* | | | | | | |
| SIF | 18 | **0.45*** | 1 | **0.57*** | **0.71*** | **0.57*** |
| SSRT | 18 | 0.15 | 1 | 0.23 | – | – |
| | | | | | | |
| *C. HIP GABA: PPI* | | | | | | |
| DLPFC | 18 | **−0.57*** | 0 | **−0.56*** | – | **−0.57*** |

Robust = skipped correlation using Pearson's product-moment *r*-value on data with bivariate outliers removed[53]. Out = number of bivariate outliers automatically removed[53]. Degrees of freedom on the skipped correlation = *n* − 2 − Out. Control = partial correlation controlling for Sex, levels of hippocampal glutamate (HIP Glu/Cre), and hippocampal gray matter concentration (HIP GM). Degrees of freedom on the Control partial correlation = *n* − 5− Out. Functional specificity = partial correlation controlling for Sex, HIP Glu/Cre, HIP GM, and the covariates from the Stop-Signal Task. For the HIP BOLD analyses, hippocampal BOLD response on Stop trials was used as a control for the No-Think partial correlation, and BOLD response on Go trials as a control for the Think partial correlation. For Behavior, the stop-signal response time was used as a control for the suppression-induced forgetting partial correlation. Degrees of freedom on the Functional Specificity partial correlation = *n* − 6 − Out. Anatomical specificity = partial correlation controlling for Sex, HIP Glu/Cre, HIP GM, and PFC GABA/Cre. Degrees of freedom on the Anatomical Specificity partial correlation = *n* − 6 − Out. Bold entries and asterisks indicate significance at 95% boot-strapped confidence intervals

interneurons[21]. Prior work with non-human primates, combining fMRI with cortical electrophysiology, suggests that stimulus-induced negative BOLD responses in visual cortex arise, in part, due to increases in neuronal inhibition[64]. Moreover, in humans the magnitude of task-induced negative BOLD responses in anterior cingulate have been linked with co-localized [1]H MRS estimates of GABA concentration[58, 60]. Together, these findings raise the possibility that negative BOLD responses in the hippocampus may also be linked with neuronal inhibition, and thus, co-localized [1]H MRS estimates of GABA concentration. If so, our MRS measure of baseline GABA should predict reduced memory-driven BOLD responses arising during the Think/No-Think task. In contrast, our motor action inhibition task, despite requiring focused attention and inhibitory control, should not depend on hippocampal processing, and so baseline GABA may be less related to hippocampal BOLD signal during this task. The data confirmed these expectations (Table 1A): Robust correlation analyses demonstrated that hippocampal GABA significantly predicted hippocampal BOLD response magnitude during both the Think and No-Think conditions; it did not, however, predict BOLD during either the Go or Stop conditions. Partial correlation analyses confirmed that these relationships were not driven (or masked) by participant sex, hippocampal gray matter content, or hippocampal glutamate concentrations (Table 1A).

Although the foregoing patterns suggest a functionally specific role of hippocampal GABA in memory processes, it is important to determine whether this relationship survives even when any relationship between GABA and BOLD in non-memory tasks is accounted for. This control analysis is especially necessary given that action stopping, like thought suppression, also reduced hippocampal BOLD signal (right HIP: $t_{23}$ = 2.42, $p$ = 0.02, left HIP: $t_{23}$ = 3.65, $p$ = 0.001, Fig. 1c). Stopping-related reductions in hippocampal BOLD could signify that action stopping engages mechanisms similar to thought suppression to disrupt hippocampal function; alternatively, they may simply be a passive side effect of performing a difficult task[20] (e.g., reduced afferent input to the hippocampus due to heightened task focus). In the former case, BOLD responses induced by both No-Think and Stop trials should share variance with hippocampal GABA, whereas in the

latter case, the variance explained by thought suppression (active inhibition) should differ from that explained by action stopping (task difficulty) due to their differing mechanistic origins. To distinguish these two alternatives, we conducted a partial correlation analysis on the relationship between hippocampal GABA and BOLD response during No-Think trials that additionally controlled for BOLD response during Stop trials (Table 1A). We found that the relationship between hippocampal GABA and BOLD response during No-Think trials persisted even when controlling for Stop-induced BOLD response. We obtained a similar finding when we performed this analysis using Think and Go trials. These patterns suggest that BOLD response reductions during motor stopping likely have a different mechanistic origin, perhaps relating to task difficulty. They also confirm the functional specificity of hippocampal GABA/BOLD coupling to memory processes in the context of the Think/No-Think task: only memory task-related signals to the hippocampus drove changes in BOLD signal amplitude that scaled with resting concentrations of hippocampal GABA, such that the higher the hippocampal GABA, the lower the observed BOLD response during memory retrieval and memory suppression.

We next considered the possibility that the relationships between hippocampal GABA and BOLD measures were not specific to hippocampal GABA. This relationship could, for example, reflect GABAergic integrity throughout the broader fronto-hippocampal pathway supporting the suppression of unwanted thoughts[28–35]. If so, hippocampal BOLD responses should share variance with both hippocampal and DLPFC GABA concentrations. Alternatively, if the relationship is anatomically specific, hippocampal GABA should share unique variance with hippocampal BOLD responses. To distinguish these alternatives, we conducted a partial correlation analysis on the relationship between hippocampal GABA and BOLD responses during No-Think trials, additionally controlling for DLPFC GABA (Table 1A). Consistent with anatomical specificity, the relationship between hippocampal GABA and No-Think-induced BOLD response persisted, even when controlling for DLPFC GABA concentration. Anatomical specificity also held for the partial correlation between hippocampal GABA and BOLD responses during Think trials, controlling for DLPFC GABA. Finally, we examined whether DLPFC BOLD responses during the No-Think and Think conditions were correlated with DLPFC GABA. No relationships were observed, even when controlling for participant sex, DLPFC glutamate, and DLPFC gray matter concentrations (Table 2A). Moreover, whereas robust correlations on the visual cortical control ROI revealed a negative correlation between visual cortical GABA and co-localized BOLD during Think and No-Think trials, these relationships did not survive after controlling for participant sex, visual cortical glutamate, and visual cortical gray matter concentrations (Table 2D). Together, these findings suggest that hippocampal GABA is not simply a proxy for brain-wide GABA integrity, but rather captures region-specific variation, and that this variation is distinctively related to co-localized BOLD responses during memory retrieval and memory suppression.

The foregoing findings provide evidence for a functionally and anatomically specific relationship of GABA to memory tasks, whereby higher hippocampal GABA predicts reduced BOLD signal. Interestingly, we observed this relationship for both suppression and retrieval. Although we did not anticipate that hippocampal GABA would exhibit a negative relationship with retrieval-induced upregulation of hippocampal BOLD, this observation can be understood in retrospect. A key observation is that [1]H MRS indices of bulk tissue GABA are unlikely to be tied to BOLD signal in any single task, but rather should be related to any psychological process that evokes high demand on local

**Table 2 Intermodal relationships of DLPFC and visual cortical GABA**

| Intermodal relationship | N | Robust | Out | Control |
|---|---|---|---|---|
| *A. PFC GABA: PFC BOLD* | | | | |
| No-Think | 23 | −0.02 | 1 | 0.26 |
| Think | 23 | −0.05 | 1 | 0.27 |
| *B. PFC GABA: behavior* | | | | |
| SIF | 23 | −0.04 | 1 | −0.23 |
| *C. PFC GABA: PPI* | | | | |
| DLPFC | 23 | 0.04 | 0 | −0.20 |
| *D. VIS GABA: VIS BOLD* | | | | |
| No-Think | 20 | **−0.42*** | 1 | −0.31 |
| Think | 20 | **−0.43*** | 0 | −0.31 |
| *E. VIS GABA: Behavior* | | | | |
| SIF | 20 | 0.32 | 3 | 0.25 |
| *F. VIS GABA: PPI* | | | | |
| DLPFC | 20 | 0.12 | 0 | 0.27 |

Robust = skipped correlation using Pearson product-moment *r*-value on data with bivariate outliers removed[53]. Out = number of bivariate outliers removed[53]. Degrees of freedom on the skipped correlation (*n* − 2 − Out). Control = partial correlation controlling for Sex, levels of PFC or VIS Glutamate (Glu/Cre), and PFC or VIS gray matter concentration (GM). Degrees of freedom on the Control partial correlation (*n* − 5 − Out). Bold entries and asterisks indicate significance at 95% boot-strapped confidence intervals

GABAergic interneuron populations. Retrieval processes during the Think condition likely also evoke increases in GABAergic interneuron activity. It is widely known, for example, that rhythmic firing of GABAergic interneurons in the hippocampus makes an essential contribution to the theta rhythm, which is believed to be critical for encoding and retrieval[65–68]. One speculation is that the observed relationship between GABA and retrieval-related BOLD signal reflects this key role of GABA interneurons, a possibility consistent with the fact that theta activity is sometimes associated with reduced BOLD signal (though the relationship of these variables is complex)[69]. This potential rhythmic engagement of hippocampal GABAergic interneurons in our retrieval task cannot be evaluated in the present data. Importantly, however, this speculated role of GABAergic interneurons in retrieval is functionally distinct from the increases in tonic inhibition[36, 37] that we had hypothesized might underlie retrieval suppression, and the associated reduction in hippocampal BOLD signal.

Although the foregoing relationships between GABA and BOLD cannot, by themselves, distinguish the hypothesized tonic inhibition mechanism, our behavioral measures provide important information relevant to a distinct role of inhibition during thought suppression. If suppression engages GABAergic interneurons in a distinct manner, as we have hypothesized, baseline GABA measures should predict how effectively participants forget the thoughts they try to suppress. We tested this possibility by relating hippocampal GABA concentrations to performance on the final surprise recall test of the Think/No-Think paradigm (Methods section, Table 1B). Consistent with our initial hypothesis, a robust correlation analysis revealed that participants with higher hippocampal GABA exhibited better suppression of unwanted content, as reflected in higher suppression-induced forgetting (Baseline–No-Think). Hippocampal GABA did not, in contrast, predict retrieval-induced facilitation (*r* = −0.12, 95% CI: (−0.78 to 0.51)). We also did not observe a relationship between hippocampal GABA and pre-scan recall performance of the studied word pairs (*r* = −0.10, 95% CI: (−0.50 to 0.41)), indicating that the positive relationship between hippocampal GABA and

suppression-induced forgetting is unlikely to be explained by a relationship between GABA and baseline learning success. To further interrogate the functional specificity of the relationship of hippocampal GABA to thought suppression, we examined whether hippocampal GABA predicted general indices of inhibitory control ability, as assessed with motor action-stopping speed on the SS Task (the stop signal reaction time). No such relationship was detected (Table 1B). Control partial correlation analyses confirmed that these relationships were not masked by participant sex, hippocampal gray matter content, or hippocampal glutamate concentrations. Together, these findings point to a specific relationship of hippocampal GABA to thought suppression, and not to general inhibitory control ability. Indeed, even when we accounted for individual variation in general inhibitory control ability (by including stop signal reaction time as a covariate in partial correlations), the hippocampal GABA-forgetting relationship was, if anything, strengthened (Table 1B).

We next tested whether suppression-induced forgetting was uniquely predicted by hippocampal GABA, or, was instead related to brain-wide GABA concentrations indexed from our three [1]H MRS ROIs. To evaluate anatomical specificity, we conducted a partial correlation analysis that examined the relationship between hippocampal GABA and suppression-induced forgetting, while controlling for shared variance with DLPFC GABA. Critically, we found that the relationship between hippocampal GABA and suppression-induced forgetting persisted in this model (Table 1C). Moreover, we also directly tested whether DLPFC GABA itself predicted suppression-induced forgetting. We observed no such relationship, even when controlling for participant sex, DLPFC glutamate, and DLPFC gray matter concentration (Table 2B). GABA concentrations outside of the fronto-hippocampal pathway, in the visual cortical control ROI, also failed to account for significant variance in suppression-induced forgetting (Table 2E).

The foregoing findings suggest that GABA concentrations local to the hippocampus contribute to the persisting disruption of intrusive thoughts in healthy participants. Although the cellular mechanisms underlying the influence of GABA on memory cannot be established from MRS data, increased tonic inhibition has, in animal models, been found to attenuate synaptic plasticity, impairing memory[70, 71]. Conversely, in humans, experimentally reducing local GABA concentrations in motor cortex facilitates motor plasticity and increases co-localized BOLD response[72]. Taken together, these findings suggest that a suppression-related increase in tonic GABAergic inhibition could, in principle, disrupt plasticity in the hippocampus underlying episodic retention. More broadly, however, these findings are consistent with the hypothesis that although both retrieval and suppression are likely to engage hippocampal GABAergic inhibition networks, they do so in functionally distinct ways.

**Reduced hippocampal GABA compromises prefrontal control.** If intentionally suppressing thoughts engages hippocampal GABAergic networks in a functionally distinct manner, some mechanism must drive this activity. Prior effective connectivity analyses indicate that suppressing retrieval involves a goal-related signal that originates in right DLPFC and spreads downstream, via polysynaptic pathways[28], to the hippocampus, integrating these regions in a task-dependent manner[28–35, 41]. If this fronto-hippocampal pathway provides afferent input that drives GABAergic processes during suppression, then how strongly DLPFC and hippocampus functionally integrate should depend on the availability of hippocampal GABA to implement retrieval stopping. Specifically, higher concentrations of hippocampal GABA should predict stronger negative DLPFC-hippocampal

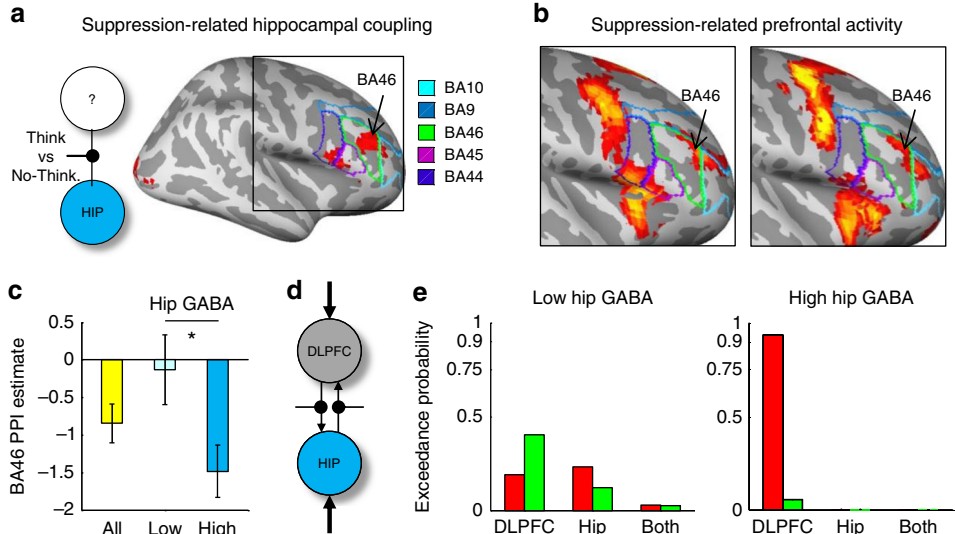

**Fig. 3** Hippocampal GABA predicts DLPFC-Hippocampal connectivity during thought suppression. **a** Schematic of psychophysiological interaction analysis (PPI) with hippocampal (HIP) seed and conditions modulating HIP connectivity. Significant PPI effects arose in right lateral prefrontal cortex: Brodmann's area (BA) 46/9 (DLPFC) and BA45 (VLPFC), displayed as colored boundaries (see legend). **b** DLPFC activity during suppression (No-Think > Think) in the current (left) and in a prior study[31] (right) overlapped with the PPI effects. **c** Functional connectivity: Suppression negatively modulated fronto-hippocampal coupling (PPI estimate, *y* axis), with the strength of negative coupling differing between low- and high-GABA subgroups (Independent samples *t*-test, *\*p* < 0.05). Error bars represent standard error of the mean. **d** The six bidirectional dynamic causal models of the DLPFC–HIP network varied according to two parameters: Which Task modulated connectivity (horizontal lines: No-Think or Think) and source of Driving Input (Outer arrows: DLPFC, HIP, or BOTH). **e** Effective connectivity: for Low GABA participants, no clear evidence for a role of DLPFC in modulating connectivity emerged in any model. For high hippocampal GABA participants, model evidence (exceedence probabilities) favored a model with inputs to DLPFC driving the network and the No-Think task modulating connectivity

coupling. To test this possibility, we first used psychophysiological interaction (PPI) analysis[67] to examine brain-wide task-dependent connectivity with the hippocampus, isolating all regions with which it shows suppression-related coupling (Supplementary Methods). The PPI thus enabled a whole-brain (data-driven) search for patterns of covariance with the hippocampus that differed significantly depending on whether participants retrieved thoughts (Think) or suppressed them (No-Think), after accounting for variance explained by main effects of task (No-Think, Think, Go, and Stop) and physiological (task-independent) correlations with the hippocampus (Fig. 3a).

We observed task-dependent connectivity between the hippocampus and the right DLPFC ($t_{23}$ = 3.58, $p$ = 0.034 after small volume FWE correction with an a priori DLPFC ROI[31]; Fig. 3a). Additional activations were detected in right inferior frontal gyrus, and in early visual cortex (at a more liberal uncorrected threshold, $p$ < 0.005). This connectivity effect thus showed high anatomical specificity. We next projected onto the same cortical surface (A) the whole-brain main effect contrasts of retrieval suppression [No-Think > Think] observed in the current study, and (B) the study from which our a priori ROI is derived[31]. These clusters overlapped at the juncture of Brodmann Area (BA) 46, 9, and 10 in right middle frontal gyrus (Fig. 3b). Critically, these functional connectivity effects reflect negative modulation, that is, a task-dependent inversion of BOLD activity in hippocampus relative to DLPFC, consistent with our hypothesis that suppression-induced recruitment of right DLPFC signals retrieval suppression—and hence downregulation of BOLD activity—in the hippocampus (Fig. 3c). No regions expressing task-dependent positive modulation with the hippocampus were detected. These initial connectivity findings confirm that suppressing unwanted thoughts functionally integrates the right DLPFC and the hippocampus, consistent with a possible role of DLPFC in modulating unwanted hippocampal retrieval activity.

Of central interest, however, is whether the negative coupling observed between the DLPFC and hippocampus during suppression was associated with hippocampal GABA concentrations, as would be expected if local GABA contributed to inhibitory control. To assess this, we first tested with robust correlation analysis whether indices of DLPFC–hippocampal connectivity varied continuously with hippocampal GABA. Consistent with our hypothesis, individuals with higher hippocampal GABA exhibited stronger negative coupling of the DLPFC with the hippocampus (see Table 1C). Control partial correlation analyses confirmed that this relationship was not driven by variation in participant sex, hippocampal gray matter content, or hippocampal glutamate concentrations. The relationship also showed striking anatomical specificity within the fronto-hippocampal control pathway: A partial correlation analysis, controlling for DLPFC GABA, revealed that hippocampal GABA uniquely predicted PPI indices of connectivity with the DLPFC (Table 2C). We also examined whether DLPFC GABA was itself correlated with PPI indices of DLPFC connectivity. We observed no such relationship, even when controlling for participant's sex, DLPFC glutamate, and DLPFC gray matter concentration (Table 2C). GABA concentrations in the visual cortical control ROI also failed to account for significant variance on the PPI indices of fronto-hippocampal coupling (Table 2F).

To further explore the anatomical specificity of hippocampal GABA to DLPFC-hippocampal coupling, we median split our sample into two subgroups with lower and higher hippocampal GABA concentrations ($t_{16}$ = 6.10, $p$ = 0.00002). Crucially, these subgroups were matched on DLPFC ($t_{16}$ = 1.25, $p$ = 0.23) and visual cortical GABA ($t$ < 1), as well as on age, sex and several cognitive measures, including performance during initial word-pair training, motor response speed, and motor action inhibition (Supplementary Table 1). This approach enabled us to determine whether connectivity patterns differed depending on local

hippocampal GABA, independent of GABA in other regions. We found that task-dependent DLPFC-hippocampal connectivity differed significantly between the high and low hippocampal GABA subgroups ($t_{16} = 2.39$, $p = 0.03$): Whereas the high GABA subgroup showed negative coupling during retrieval suppression, the low GABA subgroup did not (Fig. 3c). Taken together, these findings are consistent with the hypothesis that suppressing unwanted thoughts engages GABAergic interneurons local to the hippocampus in a functionally distinct manner to implement an inhibitory control signal driven by DLPFC.

Although PPI analysis shows that condition-dependent coupling occurs between the DLPFC and the hippocampus, one cannot infer that suppression causes this integration or that the input driving hippocampal suppression originates in the DLPFC. To identify the causal dynamics of the proposed network, we used dynamic causal modeling, a Bayesian statistical framework for inferring effective connectivity between brain regions through a network composed of a small number of key brain regions[68]. We used a model space from a prior study[31] that included the DLPFC and hippocampus as key regions. Briefly, the model space was defined by: (i) intrinsic bidirectional connections between the right DLPFC and the right hippocampus (modeling regional interactions that may be mediated polysynaptically), (ii) task-induced modulation of either the top-down connections from DLPFC to the hippocampus, bottom–up connections from the hippocampus to DLPFC, bidirectional connections, or no connections, and (iii) task-related input sources that drive activity in the network (e.g, No-Think and Think inputs driving activity either via the hippocampus, the DLPFC, or both). To further confirm the functional selectivity of the DLPFC-hippocampal pathway to suppressing unwanted thoughts, rather than to the broader process of inhibiting any type of response, we performed a parallel dynamic causal modeling analysis, using an analogous model space, but substituting the No-Think and Think conditions (parameters ii and iii) with the Stop and Go conditions of the stop-signal action inhibition task. We fit all of these models to the fMRI time series in each participant (Supplementary Methods).

Using this model space, we first evaluated the Think/No-Think task in the whole sample ($N = 24$). Replicating prior findings, we found the strongest evidence for models with bidirectional coupling between DLPFC and hippocampus[31, 35] (Supplementary Notes). By contrast, when we performed a parallel analysis substituting the Think/No-Think task with the stop-signal action inhibition task (Stop, Go) we found no evidence that action-stopping modulated DLPFC-hippocampal connectivity (Supplementary Notes). Thus, the effective connectivity findings for the thought suppression and action inhibition tasks accord well with the hypothesis that DLPFC-hippocampal network dynamics support a function specific to thought suppression.

Next, we tested how the availability of GABA in the hippocampus related to network architecture for the High and Low hippocampal GABA subgroups (Fig. 3d). If hippocampal GABA enables goal-directed input from the DLPFC to disrupt hippocampal retrieval processes during suppression, lower GABA concentrations should mute DLPFC influence on network dynamics. To test this hypothesis, we conducted two parallel Bayesian model analyses for the High and Low hippocampal GABA subgroups within the winning bidirectional family (Fig. 3e). Consistent with our hypothesis, the Low GABA subgroup showed little evidence that DLPFC inputs drove network activity during any condition. By contrast, the analysis in the High GABA subgroup isolated a single winning model, in which inputs to the DLPFC drove network dynamics, and the No-Think condition modulated fronto-hippocampal coupling (EP: 93%; Posterior probability: 67%). Corroborating this apparent difference in network dynamics, an ANOVA comparing

between-group estimates of model parameters (Supplementary Methods) revealed a significant interaction between Group (Low and High hippocampal GABA) and Condition (No-Think vs. Think): ($F_{1,16} = 6.23$, $p = 0.024$). Driving input into the DLPFC significantly differentiated thought suppression from retrieval (No-Think vs. Think) in the High hippocampal GABA subgroup ($t_8 = 2.9$, $p = 0.02$); the Low hippocampal GABA subgroup did not show inputs that differed by task ($t_8 < 1$). These findings indicate that GABA in the hippocampus plays a distinctive and pivotal role in enabling goal-related signals entering the DLPFC to affect network dynamics during suppression. More broadly, they integrate the foregoing multimodal imaging results—in which we linked GABA to BOLD responses in the hippocampus and behavior—into an explanatory model of fronto-hippocampal dynamics during the suppression of unwanted thoughts.

## Discussion

The ability to disengage from unwanted thoughts is essential to mental health. Our results suggest that GABAergic inhibition of hippocampal retrieval processes enables such thoughts to be suppressed. The data suggest that whereas the DLPFC contributes a top-down control signal needed for retrieval stopping, as previously shown[28–35, 41], the efficacy of this signal depends on hippocampal GABA to implement suppression. With lower hippocampal GABA concentrations, the influence of prefrontal control signals on hippocampal activity and on the later accessibility of the unwanted thought is muted, as reflected in a weaker influence of suppression on hippocampal BOLD signal, reduced forgetting of intrusive thoughts, and decreased negative coupling between right DLPFC and the hippocampus. Indeed, effective connectivity analyses supported an important role of hippocampal GABA in the integrity of this network: individuals with lower GABA showed little evidence that DLPFC modulated hippocampal activity, unlike individuals with higher GABA. Critically, we found an anatomically and functionally specific role of GABA in suppressing thoughts: unlike hippocampal GABA, GABA concentrations in the DLPFC were not related to either reduced hippocampal activity during suppression or to suppression-induced forgetting; and unlike thought suppression, behavioral indices of action stopping (a demanding inhibitory control task) were not related to reduced hippocampal activity during suppression or to hippocampal GABA concentrations. Taken together, these findings suggest that a functionally specific fronto-hippocampal inhibitory control pathway underlies the ability to suppress unwanted thoughts, and that the functional integrity of this pathway may depend on GABAergic interneuron networks local to the hippocampus.

We propose that the ability to suppress a broad spectrum of mental content depends on mechanisms that stop hippocampal retrieval processes via GABAergic inhibition. In this study, we measured this ability by asking cognitively healthy young adults to suppress the retrieval of simple verbal items. Given this approach, the current findings cannot directly address whether suppressing the more complex and aversive content that typically intrudes in many psychiatric conditions also relies on hippocampal GABA. However, when considered together with literature on retrieval suppression, this possibility seems likely. Suppression-induced forgetting occurs for a range of stimuli including neutral or unpleasant words[29–31, 38, 42], visual objects[34], neutral or unpleasant scenes[32, 33, 41, 43, 44], autobiographical memories[45], and person-specific fears about their future[35]. In all of these cases, retrieval suppression engages a right lateralized fronto-hippocampal inhibitory control pathway closely matching the one shown here. Moreover, during the suppression of aversive images[41], this fronto-hippocampal inhibitory pathway shows especially pronounced reactive engagement when the to-be-

suppressed content intrudes into awareness, consistent with a role in suppressing unpleasant and intrusive content. Indices of retrieval suppression ability also predict trait anxiety[48], post-traumatic stress symptoms[44], rumination[50], and self-reports of thought control ability[44, 51]. Together, these observations suggest that a GABAergic hippocampal mechanism suppresses retrieval over a broad spectrum of perseverative thoughts (whether images, episodes, or worries about future events). This proposed mechanism linking hippocampal GABA to the volitional control over the contents of awareness may help to interpret a growing body of human[10, 13–20] and animal[21, 22, 24–27] research pointing to hippocampal GABAergic hypofunction as a pathophysiological driver of intrusive symptoms. Ultimately, however, determining whether successful thought suppression relies on local hippocampal GABA requires a direct test of this generalization, together with experimental manipulations of GABA rather than the individual differences correlational approach used here.

Our findings raise questions about the cellular and local circuit mechanisms through which hippocampal GABA enables the suppression of unwanted thoughts. Inferences about neural mechanisms are necessarily limited because $^1$H MRS only provides one estimate of GABA for each large region of interest (for example the hippocampus or the DLPFC), and this estimate reflects a combination of intracellular, synaptic, and extrasynaptic GABA from all types of GABAergic interneurons in that ROI[73]. As a working hypothesis, however, we suggest that retrieval suppression may arise in part from an increase in tonic inhibition of principal cells in the hippocampus caused by sustained disinhibition of local GABAergic interneurons. Hippocampal inhibitory interneurons (which are exclusively GABAergic) undergo rhythmic inhibition from GABAergic pacemaker cells projecting from the medial septal nucleus of the basal forebrain[74,75]. These septo-hippocampal inputs, together with hippocampo-septal back-projections, contribute to driving theta oscillatory activity widely considered essential for encoding and retrieval[75]. Lesions or inactivation of the medial septal nucleus desynchronize hippocampal rhythms, reduce overall EEG amplitude, abolish hippocampal theta, and impair episodic memory[76]. These outcomes likely arise in part because disrupting the medial septal nucleus eliminates inhibitory septo-hippocampal inputs to the hippocampus, disinhibiting hippocampal GABAergic interneurons, increasing the tonic inhibition they exert on principal cells[77]. For these reasons, it has been hypothesized that inhibiting medial septal nucleus activity provides a means of suppressing hippocampal mnemonic processes so that unwanted information can be disregarded[36, 37]. Supporting this proposal, the suppression of unwanted thought was also found to downregulate activity in the medial septal nucleus (Supplementary Methods and Supplementary Fig. 2), possibly indicating disruption of its input to the hippocampus. How these regions interact during retrieval suppression remains an open question. However, this possibility converges with prior findings showing that retrieval suppression reduces theta-power in the medial temporal lobes[78] and also broadly disrupts memory for all recent events, the retention of which depends on the hippocampus[79]. The present findings, therefore, may indicate in humans that task-induced suppression of the medial septal nucleus can disrupt mnemonic functions in the hippocampus. If so, we suggest that the putative influence of DLPFC on hippocampal GABAergic activity may include a signal that suppresses pacemaker cells in the medial septal nucleus, triggering tonic inhibition of principal cells. This pacemaker suppression hypothesis should be a focus of future work.

The current study did not seek to isolate the polysynaptic pathway through which DLPFC suppresses activity in the medial septal nucleus or in the hippocampus. Rather, our findings underscore the high-level function of hippocampal GABA in integrating the prefrontal cortex and hippocampus into an effective inhibitory control pathway. This function may provide a unifying account of several key observations in neurobiological research on psychiatric disorders. Reduced functional connectivity between the prefrontal cortex and the hippocampus is increasingly recognized as a core pathophysiological feature shared by a range of psychiatric disorders, and the neural mechanisms of such fronto-hippocampal disconnection have been examined in animal models[80–82]. However, many of the disorders exhibiting fronto-hippocampal disconnection also are characterized by the hippocampal hyperactivity and intrusive symptomatology of main interest in the present study[10, 14, 16–20]. The present findings raise the possibility that GABAergic disinhibition in the hippocampus contributes to reduced fronto-hippocampal connectivity, hippocampal hyperactivity, and intrusive symptomatology. These differing symptoms may therefore represent a common dysfunction of the fronto-hippocampal inhibitory control pathway specified in the present study. This hypothesis suggests that estimates of hippocampal GABA should be related to hippocampal hyperactivity and to reduced resting state connectivity between the hippocampus and the prefrontal cortex; it may also partially account for the widely established difficulty in suppressing default mode network activity arising in a range of psychiatric disorders characterized by intrusive symptomatology[20]. The current work therefore provides a neurobiological framework for studying psychiatric disorders that share persistent intrusive thoughts as a common symptom. More broadly, in bridging molecular neuroscience and higher-level cognition, these findings lay the groundwork for a multi-level model system of inhibitory control over thought.

## Methods

**Participants**. Thirty right-handed native English speakers (seven males) aged between 19 and 36 years (Mean = 24.7, SD = 4.3) were paid to participate. Prior to experimental procedures, participants were instructed to refrain from alcohol or psycho-active drugs in the 24 h period prior to the scan. Five participants were excluded for not reaching the learning criterion of 40% on the Think/No-Think task. One participant was excluded for drowsiness during fMRI acquisition. Participants reported no history of neurological, medical, visual, or memory disorders. The project was approved by the Cambridge Psychology Research Ethics Committee, and all participants gave written informed consent. Participants were asked not to consume psychostimulants, drugs, or alcohol before the experimental period.

**Behavioral tasks**. Participants performed adapted versions of the TNT[38] and SS[40] tasks, which were interleaved in a mixed block/event-related design.

SS task: We used a modified SS procedure with three phases: (1) a stimulus-response mapping phase, during which participants learned associations between color cues and responses; (2a) a practice phase, during which all participants practiced the response mapping with the occurrence of stop-signals; (2b) an extended practice phase interleaved with the TNT; (3) the critical SS task phase, during which they were scanned.

During the stimulus-response mapping phase, participants were trained in responding to four different colored circles (red, blue, green, and yellow), by pressing one of two buttons. Thus, each button had two colors assigned to it. Participants first learned to associate each color to its particular response. A fixation cross was presented in the screen for 500 ms, followed by a colored circle at the top of the screen and a cartoon depicting the response box below it, and the correct response indicated by a white arrow. This was shown until the participants pressed the correct button. After showing two colors twice, participants learned the response mapping for the two colors, during 20 trials. Subsequently, two new colors were introduced and participants learned the response mapping for the two new colors for another 20 trials. Participants then practiced the response mapping for the four colors together until they reached a total of 10 correct subsequent trials for each color. For these latter trials, feedback was provided for errors and slow responses, but without the white arrow indicating the correct responses.

Once the stimulus-response mapping was established, participants practiced the SS task. Participants were instructed that they should keep trying to respond as fast as possible to each color, but that in some of the trials a beep would sound at varying times after the circle appeared, in which case they should prevent their response and not press the button. Participants performed 96 trials of the stop-signal task, before moving on to the TNT training. Thus, in Go trials participants made their responses as fast as possible, whereas in Stop trials an auditory tone succeeded cue onset, signaling participants to suppress their responses. A Go trial

started with a fixation cross, presented on the screen for 250 ms, followed by a colored circle until response (for up to 1500 ms). After the response, there was a jittered inter-trial interval (mean ± SD: 750 ± 158.7 ms) and a new trial commenced. Stop trials were identical with the exception that a tone would sound shortly after the presentation of the circle. This tone had two different possible delays, one of 250 ms another of 300 ms. Of the total number of trials in this task, 30% were Stop-signal trials, whereas the remaining 70% were Go trials. Performance for stop trials was maintained at around 50% correct by using a staircase tracking algorithm that modified the forthcoming trial according to response feedback on the current trial. Specifically, if a participant correctly withdrew his/her response upon a stop-signal tone, 50 ms would be added to the stop signal delay of the next stop trial. Alternatively, if an incorrect response was made (that is, if the participant pressed a button on a stop trial), the following trial would have a stop signal delay that was 50 ms shorter. Note that the longer the stop signal delay is, the harder it is to withdraw from pressing the button. Again, during this phase, feedback was provided for both incorrect trials and slow responses (RT > 700 ms), in order to ensure participants responded as quickly as possible.

TNT task: Once participants where familiarized with the SS task, the TNT procedure was introduced. We used a modified TNT procedure[38] with four phases: (1) a study phase, during which participants encoded cue-memory pairs; (2a) a practice phase, during which all participants practiced retrieval suppression on filler pairs; (2b) an extended practice phase interleaved with the SS; (3) the critical suppression phase, during which they were scanned; and (4) the final test phase, during which we tested their memory.

In the study phase, participants encoded 60 critical cue-memory word pairs (e.g., BEACH-AFRICA). A third of those constituted the No-Think items, another third the Think items, and the final third served as baseline items for the final test. Assignment of words to the three conditions was counterbalanced across participants. In addition, they also memorized a further 18 filler pairs that were used for practice. The study phase had three stages. First, each pair appeared for 3.4 s with an interstimulus interval (ISI) of 600 ms. Second, participants overtly recalled the memories in response to the cues, which were shown for up to 6 s or until a response was given. After cue offset (and a 600 ms ISI), the correct memory appeared for 1 s. This procedure was repeated until participants recalled at least 40% of the critical memories (all but 5 participants succeeded within the maximum of three iterations of the list). Third, we presented each cue one more time for up to 3.3 s (ISI: 1.1 s), and without feedback, to assess which memories had been correctly learned.

During practice, all participants were trained on the direct suppression variant of the Think/No-Think task[31]. They were instructed to covertly recall memories for cues presented in green font (Think condition) but to avoid thinking of memories for cues presented in red (No-Think condition). On each trial, they were required to first read and comprehend the cue. In the Think condition, participants retrieved the associated memory as quickly as possible and kept it in mind while the cue remained onscreen. By contrast, in the No-Think condition, participants blocked out all thoughts of the associated memory without engaging in any distracting activity. Whenever a memory intruded into awareness, they were asked to "push it out of mind." A trial consisted of the presentation of a cue in the center of the screen for 3 s, followed by an ISI (mean ± SD: 2.3 s ± 1.7 s). We administered a questionnaire in the middle of the practice session to ensure comprehension of and compliance with the instructions.

Mixed SS/TNT training: Finally, before moving into the MRI scanner, participants performed an extended practice phase (part 2b of each task protocol) in order to familiarize themselves with the alternating block sequence of the two tasks. All blocks were 30 s in duration. The trial timings for both the SS and TNT tasks were identical to those used in part 2a of their respective practice phases. SS blocks: Each trial commenced with a fixation onset, followed by a colored circle cue onset. In total, there were 12 trials per SS block, with trials pseudo-randomly ordered. TNT blocks: Each trial commenced with a fixation onset, followed by a colored word cue onset. In total, there were 6 trials per TNT block, with trials pseudo-randomly ordered. In this practice phase participants performed eight blocks of each task.

**fMRI Tasks**. Participants performed 16 blocks per session (8 SS and 8 TNT) over 8 scan sessions while fMRI was acquired. The trial durations for both the SS and TNT tasks were identical to those used in part 2a and 2b of their respective practice phases. SS Blocks: A tracking algorithm varied the lag between cue onset and stop-signal tone according to each participant's performance, thereby ensuring 50% stopping success. TNT Blocks: In each session, participants saw each cue of the recall and suppress condition once. Thus, they suppressed or recalled each memory 8 times in total. In each block, No-Think and Think trials were interspersed pseudo-randomly, and the ISI was jittered (≥0.5 s; mean ± SD: 2.3 ± 1.7) to optimize the event-related design (as determined by optseq2: http://surfer.nmr.mgh.harvard.edu/optseq). The proportion of trials that were Think items was greater (58%) than the proportion of trials that were No-Think items (42%) to better resemble the higher frequency of Go trials than Stop trials during the stop-signal task. This was accomplished by inserting a greater number of Think trials on "filler" word pairs, without changing the frequency of Think trials on critical experimental items. During the ISIs, a fixation cross appeared. To minimize carry-over effects, 4 s rest periods were interspersed between blocks. Each block also began with several trials on filler items that were not scored to reduce task-set switching effects between blocks. Moreover, to limit fatigue, participants were allowed to have a break of up to

30 s in between each scan session. We administered a questionnaire after the 4th (middle) run to ensure comprehension of and compliance with the instructions.

TNT final test phase: Participants attempted to retrieve all memories, i.e., irrespective of retrieval status (No-Think, Think, and Baseline). Before the actual test took place, participants attempted to retrieve 10 items, 6 of which they had not seen since the initial study and 4 of which they had not encountered since the interleaved Stop-Signal/TNT practice phase. Participants were warned that the cue words in this phase were hints they had not seen for a long time, and instructed to think back to the first phases of the experiment. This was done in order to reinstate the context of the study phase. In the formal test, cues were presented for a maximum of 3.3 s or until a response was given (ISI: 1.1 s). A response was coded as correct if participants recalled the memory while the cue was onscreen. In a same-probe test, memory was probed with the original cues. A second, independent-probe test was used to test whether forgetting generalized to novel cues[38]. Here we cued with the semantic category of the memory and its first letter (e.g., CONTINENT-A for AFRICA). The order of these two tests was counterbalanced across participants. During debriefing, participants rated on a five-point scale for each suppress item the degree to which they had focused on the cue as it appeared on the screen. For each item, they also indicated on a five-point scale their difficulty in suppressing the memory (1: not difficult at all; 5: very difficult). Finally, a three-item compliance scale was administered to determine whether participants had followed the No-Think instructions properly, or had instead engaged in strategies that violated instructions, such as intentionally rehearsing No-Think items.

**Functional magnetic resonance imaging (fMRI)**. Scanning was performed on a 3 T Siemens Tim Trio MRI system using a 32-channel whole-head coil. Participants were positioned supine and foam pads were used to fixate the subject's head within the RF coil housing. High-resolution (1 × 1 × 1 mm) magnetization-prepared, rapid gradient echo (MP-RAGE) T1-weighted images were collected for anatomical visualization and normalization (other imaging parameters were as follows: FOV 256 × 240 × 192; TR: 2250 ms; TE: 2.99 ms; flip angle 9°). Functional data were acquired using a gradient echo, echoplanar pulse sequence (TR = 2000 ms, TE = 30 ms, 32 axial slices, descending slice acquisition, 3 × 3 × 3 mm voxel size, 0.75 mm interslice gap). The first four volumes of each session were discarded to allow for magnetic field stabilization.

Functional activation was determined from the BOLD signal using the software Statistical Parametric Mapping (SPM12, University College London, London, UK; http://www.fil.ion.ucl.ac.uk/spm/software/spm12/). Following image reconstruction, the time series data for each participant were motion corrected (translational motion parameters were less than one voxel for all included participants) and then corrected for slice acquisition temporal delay. All ROI-based analyses were performed on native space images preprocessed to this point. The random effects PPI analyses were performed in MNI space. To do so, the contrast maps for the PPI effect in each participant were first normalized using the parameters derived from the nonlinear normalization of individual gray matter T1 images to the T1 template of the Montreal Neurological Institute (MNI, Montreal), and spatially smoothed using a 8-mm FWHM Gaussian kernel for univariate analyses.

Single-subject time series data were submitted to a first-level general linear statistical model, GLM. Using the SPM design specification, the task-specific box-car stimulus functions were convolved with the canonical hemodynamic response function (HRF). Each model included session-specific grand mean scaling, high-pass filtering using a cutoff frequency set at 1/128 Hz, and the AR1 method of estimating temporal autocorrelation. Regressors were created by convolving box-car functions with a canonical HRF. To account for differences in stimulus duration between the TNT and SS tasks, the durations of the box-car functions varied according to task. For TNT trials, we modeled No-Think and Think trials as separate regressors, with trial durations of 3000 ms. For the SS task, correct Stop and correct Go trials were modeled as separate regressors, with trial durations derived from (group) mean response latency for each trial type. Error trials for the TNT (forgotten items) and SST (incorrect Stop, incorrect Go) were modeled as separate regressors. The six motion parameters produced at realignment were included in the model to account for linear residual motion artefacts. Percent signal change was extracted from the HIP, DLPFC, and VIS ROIs using Marsbar (http://marsbar.sourceforge.net/).

**$^1$H MRS**. The medial temporal lobes pose a methodological challenge for $^1$H MRS due to low signal-to-noise ratios. We therefore scanned all ROIs with an adapted point-resolved two-dimensional J-resolved PRESS (2D J-PRESS) MRS sequence[59], which provides major advantages over conventional 1D MRS for imaging brain areas under these circumstances[58]. The hardware configuration was identical to the fMRI scan. Participants were positioned supine and foam pads and a chin strap were used to fixate the subject's head within the RF coil housing. High-resolution (1 × 1 × 1 mm) MP-RAGE T1-weighted images were acquired (TR/TE/TI = 2000/3.53/1100 ms; FOV = 256 × 256 × 224 mm) to facilitate accurate MRS voxel positioning and for post hoc within-MRS voxel tissue-type segmentation. $^1$H MRS and fMRI data were acquired on separate days to minimize participant fatigue.

2D J-resolved $^1$H MRS data were acquired for three brain ROIs using the sequence and methodology previously described in Prescot and Renshaw[57]. The imaging parameters were identical for all ROIs: TR/TE = 2000/31-229ms, ΔTE = 2ms (100 TE steps), 4 signal averages per TE step with online averaging, 2D spectral width =

2000 × 500 Hz, and 2D matrix size = 1024 × 100, yielding a total acquisition time of 13 min 28 s. Within-ROI B0 shimming was achieved using a manufacturer-supplied automated phase map procedure in combination with interactive manual shimming. Data was only collected for a given ROI if a full-width at half-maximum (FHWM) of ≤ 24 Hz was observed for the real component of the unsuppressed water signal. Outer-volume suppression (OVS) was achieved using six saturation bands positioned at least 1 cm away from the MRS voxel faces (Supplementary Methods).

Post-scan, we applied the ProFit algorithm[83] identically to all 2D $^1$H MRS data using the supplied 2D basis set generated without considering the effects of spatial localization. Before the 2D fast Fourier transformation (FFT), we zero-filled the raw 2D matrix to 200 points along the indirectly detected (J)-dimension. The basis set comprised of nineteen metabolites (Supplementary Methods) including creatine (Cre), GABA, and glutamate. For both GABA and Glutamate, the basis functions provided by ProFit model all of the multiplets produced by each metabolite. We expressed all metabolite concentrations as a ratio to the reference Cre metabolite concentration. Structural MP-RAGE scans, acquired in the same session as $^1$H MRS, were tissue segmented to obtain measures of within-voxel gray matter (GM), white matter (WM) and cerebrospinal fluid (CSF) content for each subject (Supplementary Methods).

We undertook several post-scan steps to ensure the quality of the 2D $^1$H MRS data. First, to ensure our data reflected good magnetic resonance field homogeneity (shims), we discarded 2D spectra whose line widths deviated by more than 3 standard deviations (±3 SD) from the mean of a given voxel[84]. On the basis of these criteria, we retained 75% (18/24) of the participants for the hippocampus ROI (mean line width = 9.2 Hz ± 1.5 SD), 100% (24/24) for the DLPFC ROI (mean line width: 7.0 Hz ± 0.68 SD), and 96% (23/24) for the visual cortex ROI (mean line width: 6.9 Hz ± 0.45 SD). Second, T.W.S. and M.M.C. conducted parallel analyses of the 1D and 2D spectra to assess water suppression and contamination of spectra by macromolecules. Four ROIs were jointly identified as lipid contaminated by visual inspection of 2D spectra and post-fitting residual plots (3 in visual cortex, 1 in DLPFC), and removed from analyses. Finally, we considered variations in data quality explicitly in control analyses that assessed the impact of two potential limitations to our $^1$H MRS data: (1) lower field homogeneity of the hippocampus voxel (compared to the other ROIs), and (2) the variable interval between fMRI and $^1$H MRS acquisitions (interval range = 1–111 days across participants, interval mean ± SD = 26 ± 34 days, though notably, longitudinal $^1$H MRS indices of GABA are reliable within cognitively healthy young adults at much longer mean intervals, e.g., 225 ± 42 days[85]). To do so, we re-analyzed all primary findings (summarized in Table 1) with weighed least squares regression (WLSR), which gives each data point an amount of influence over the parameter estimates proportionate to its 'quality' (Supplementary Notes). For WLSR models assessing the field homogeneity of hippocampal data, we weighted the quality of the data according to the line widths (Hz) for the hippocampal voxel. Separate models also used the Cramér-Rao lower bound value for hippocampal GABA as a weight, providing an estimate of the error associated with model fitting. For WLRS models assessing how the interval between scans affected the data, we weighted data quality according to the number of days between sessions (reflecting the assumption that longer intervals may equal lower quality). In all cases the relationships identified in the WLRS models closely approximated the original relationships reported in Table 1, indicating that variation in field homogeneity and inter-scan interval did not substantially impact our inferences.

**Data availability**. fMRI and $^1$H MRS data acquired for this study are available via data request at MRC Cognition & Brain Sciences Unit, University of Cambridge (info@mrc-cbu.cam.ac.uk).

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

## Acknowledgements

We thank Ian Charest, Pierre Gagnepain, Richard Henson, John Duncan, James Rowe, Nikolaus Kriegeskorte, Helen Barbas, Susan Whitfield-Gabrieli, Pedro Bekinschtein, Jonathan Fawcett, and Yuhua Guo for advice and assistance. This work was supported by UK Medical Research Council grant MC-A060-5PR00 awarded to M.C.A.

## Author contributions

T.W.S., C.S.F. and M.C.A. designed the experiment, with important contributions by M. M.C. and A.P.P. T.W.S., C.S.F. and M.M.C. conducted the experiment. T.W.S. and M.M. C. analyzed the data. All authors contributed to the analysis approach and to data interpretation. T.W.S. and M.C.A. wrote the manuscript.

## Additional information

**Competing interests:** The authors declare no competing financial interests.

