## [Peer Review File · Nature Communications]

Reviewers' comments:

Reviewer #1 (Remarks to the Author):

This paper uses functional MRI (fMRI) and Magnetic Resonance Spectroscopy (MRS) to study the brain mechanisms underlying control over unwanted thoughts. Many tests were made so as to infer that these mechanisms were specific both to the brain regions involved (e.g. hippocampus rather than primary motor/visual cortex) and what was being controlled (e.g. thoughts rather than actions). I find the paper to be impressive in the quality with which a broad range of techniques have been used and harnessed together to answer an important question (how does the brain suppress unwanted thoughts ?) - a question of relevance to many psychiatric disorders.

In what follows I will review each section of the results, covering the main findings, and focussing on methodology:

(1). Thought suppression engages a functionally specific hippocampal pathway

The GLM-based mass univariate analysis of the fMRI data (Fig 1) used a correction for multiple comparisons of cluster-level inferences using high cluster forming thresholds ($p < 0.001$) - see Fig 3. This is the correct use of the technique (c.f. recent controversy over the use of cluster level inferences in fMRI; Eklund et al. PNAS, 2015).

Subjects performed a Think/No-Think task and a contrast of Think versus No-think identified left and right hippocampus, whereas a contrast Go versus Stop identified left and right M1. I'm not especially familiar with this literature - I expect the Go-versus Stop paradigm has been scanned many times using fMRI with similar results - is this correct ? Whereas, is this the first time Think/No-think has been scanned ? Additionally, DLPFC was activated during suppression of thoughts or actions.

(2) Hippocampal GABA predicts (i) reduced BOLD and (ii) successful thought suppression

(i) Hippocampal GABA predicted hippocampal BOLD response during Think and No-Think tasks (more GABA, less BOLD) but not during Go or No-Go tasks (Actions). DLPFC and visual cortical GABA did not make these predictions. These inferences were made using correlations over subjects with bootstrapped confidence intervals.

(ii) Hippocampal GABA predicted 'suppression induced forgetting' (impairment of later memory for suppressed items). Again inferences were made using correlations over subjects with bootstrapped confidence intervals. Looks fine.

(3) Reduced hippocampal GABA compromises fronto-hippocampal network dynamics.

Think/No-think tasks modulated the (undirected) connectivity between hippocampus and DLPFC. Here DLPFC was found, and this inference made, using a whole brain search using the method known as Psycho-Physiological Interaction (PPI). Here the statistical threshold was not set using a whole brain correction, but a region of interest centred on the DLPFC (Fig 4a). This seems fine given its expected role (from work prior to this paper) in behavioural inhibition.

To test for directed changes in connectivity the authors then used Dynamic Causal Modelling (Fig

4d,e). Subjects were split into those with high versus low hippocampal GABA. For higher hippocampal GABA subjects, the best network model was one in which DLPFC provided input and "No-Think" task modulating connectivity between DLPFC and hippocampus.

This was not the case for the low GABA group.

This inference was made using the 'exceedence probability' measure - indicating which (of the tested) models was the most likely (frequently used) in the population from which the subjects were drawn. Again, the application of this methodology is sound.

SUMMARY

Overall, the findings suggest that GABAergic inhibition local to the hippocampus implements prefrontal control over intrusive thoughts. This finding is consistent with previous literature (e.g. reductions of the BOLD signal in hippocampus) but the additional use of MRS with fMRI nails this down to GABA. The data analyses have been conducted in an exemplary manner and clearly support the findings.

Reviewer #2 (Remarks to the Author):

30 young adults were recruited to the study. They performed a fMRI session, where they performed a Think/No-Think task, and the Stop Signal (SS) task, to investigate the relationship between GABAergic activity in the hippocampus and inhibitory control of unwanted thoughts. MRS data were acquired in a separate session from the right hippocampus, the right DLPFC and the visual cortex. The authors showed that hippocampal GABA was inversely related to BOLD signal suppression in the hippocampus in response to thought suppression. Appropriate controls were performed.

This is an interesting study, which uses complex methodology, and was clearly performed with care and thought. The manuscript is well-written and guides the reader through the data well. However, I am somewhat unconvinced by the specificity of the results, as discussed in detail below. In addition, MRS of the hippocampus is difficult and while the authors acknowledge this and have provided some data to reassure the reader of the quality of their spectra it is currently not possible to assess the data quality fully here, making it difficult to know how to interpret their results.

Major Points

1. The authors introduce the paper in terms of a number of psychiatric conditions and then test their hypotheses on healthy controls. It is not immediately clear to me that the mechanisms underlying the inhibition of intrusive thoughts in psychiatric disorders are the same as the mechanisms underlying the instructed inhibition of thoughts in healthy controls.
2. Quantification of GABA from the hippocampus is difficult. I am reassured by the line-width reliability across the 3 voxels, but it would be useful to have some values for the fit for the GABA per se for all three voxels to determine the reliability of the measures.
3. I am not convinced that the hippocampal GABA measure is "specific". The authors say that other "difficult non-memory tasks sometimes also reduce hippocampal activity" but this was not the case with the control task here. I do not think that this can be therefore claimed to be specific to the task in question. This issue should either be addressed in the discussion directly, and the interpretation amended accordingly, or a control experiment performed.
4. The hippocampal GABA levels and the task data were acquired on 2 different days. What assurance can the authors give that either of these measures is sufficiently stable across time to make this an appropriate analysis approach. This is a particularly important question given the

gender split – GABA is thought (though not definitively shown) to vary with the menstrual cycle. Was time of day controlled for? Stimulants? Sleep? Alcohol intake the previous night?

5. If I understand correctly the subjects were trained on the tasks prior to any of the imaging. Could it not therefore be the case that the levels of hippocampal GABA here reflect how well subjects were able to learn how to perform this task, rather than reflecting the ability to inhibit thoughts per se?

6. There were relationships between hippocampal GABA and BOLD signal here and elsewhere, but the BOLD signal is complex and not well understood. Were any behavioural relationships demonstrated, either with GABA or BOLD? This would be very useful to understand the importance of this relationship.

7. The MRS methods are not given in the main body of the manuscript. Given the detail in which the behavioural and fMRI acquisitions are described this seems like an odd omission from the main text, particularly as the behavioural and fMRI acquisition and analysis is relatively standard while the MRS is certainly not.

Minor Points

1. I am not sure that the bins in the histograms in figure 2 are informative – smaller bins would give a more detailed distribution that would be more informative to the reader.

2. Many readers will not be familiar with 2D MRS – it would be useful if the authors could expand the figure legend to figure 2 to explain the figures to the non-expert.

Reviewer #3 (Remarks to the Author):

The authors present intriguing evidence suggesting an association between hippocampal GABA content and suppression of memory retrieval in a paired associates task. The manuscript has many strengths, including the memory suppression paradigm, the fMRI approach, and the neural circuit models. In addition, the approach to measuring hippocampal GABA is commendable and uncommon thus far in the literature. However, there are significant problems with the overall conceptual framework and with the approach to correlational analyses used in support of the principal aims, as well as some concerns about the GABA measures. Until these issues are addressed, it is difficult to assess the overall impact of the work.

Conceptual Framework

The authors present potentially important evidence suggesting an association between hippocampal GABA content and suppression of memory retrieval in a paired associates task. However, the conceptual framework offered in the introduction and discussion focuses primarily on cognitive and clinical phenomena that lack a clear relationship to suppression of paired associate retrieval. Word retrieval is generalized here to represent “thinking,” “thought,” “intrusive thoughts” “intrusive symptomatology” and “awareness.” Relatedly, retrieval suppression is conceptualized as “thought suppression,” “suppression of intrusive memories,” and “control of awareness.” The Oxford dictionary’s first definition of thought is “An idea or opinion produced by thinking, or occurring suddenly in the mind.” It is true that paired associated word retrieval could be considered a simple subtype of thinking, but it is not generally considered a valid proxy for the complex, clinically relevant thought processes discussed at length in the paper. For example, the words “thought,” “think” or “thinking” are found 155 times in the manuscript, but only once in the list of references.

The HC BOLD and GABA findings pertaining to retrieval and retrieval suppression are important on their own. However, these findings do not permit generalization to cognitively and phenomenologically distinct and more complex processes such as pathological worry, rumination, obsession and hallucination.

A second type of conceptual error here is presenting the GABA differences as causal, when the findings are only correlational. The authors attribute a causal role to bulk measures of HC GABA (as measured by JPRESS) when they say "GABA enables," "depends on GABA", "GABA alters", "low GABA compromises", "GABA influences." In fact, the study provides evidence for associations with bulk measures of HC GABA. Speculations about causal relationships should be minimized and clearly framed as speculation or hypotheses for future testing.

The manuscript's title incorporates both of these misleading conceptual frames, using the terms "GABA enables" and "unwanted thoughts." In contrast, the study actually shows evidence that HC GABA is associated with volitional memory suppression.

Overstating and overgeneralizing the findings occurs in many places in the text. For example, line 10 states "In so doing, we isolate a fundamental mechanism enabling inhibitory control over thought: GABAergic inhibition of hippocampal activity." "Isolate a fundamental mechanism" is much too strong a phrase, "enabling" is speculative, and "control over thought" is much too general to associate with HC GABA based on this study.

Correlations

Line 202 states "Because the robust and partial correlation analyses yielded similar conclusions, we focus on the partial correlations for simplicity."

The reader assumes that a study principally aiming to examine the association between HC BOLD and HC GABA would have an a priori statistical plan for testing this association. If so, which of these two approaches to correlation analysis was chosen a priori? All results should be reported using the a priori method, with secondary comments on the convergence or divergence of results found with an alternate method.

As written, there is a confusing intermixing of robust and partial correlation approaches. For example, line 202 suggests that the results of the partial correlation analyses are presented in the main paper. However, line 214 indicates that CI are used for testing significance and cites the papers on robust correlations. This suggests that robust correlations are being reported for these comparisons. Again on line 353, the citations for robust correlations are given in a context where they appear to be reporting partial correlations.

In addition, there is a lack of consistency in how correlations are applied in the manuscript. Sometimes the authors provide direct comparisons between correlations, and sometimes they don't. For example, the authors report that HC BOLD-GABA correlations are significant during memory task components and not significant during motor task components. They interpret this as a selective finding, but they omit direct comparison of the correlations across tasks. However, the authors include a direct comparison between correlations for a different contrast on lines 241-244. Sometimes they include the GO condition BOLD responses as covariates in relevant analyses (line 240-1) and sometimes they don't (lines 224). The result is the appearance of selectively focusing on findings that support their model and not making sincere attempts to challenge or disprove the model. The relatively low power of the key contrasts (N=18) may have a role in this selective reporting.

What type of robust correlation was used? Was it bend, skipped, or some other? If bend, what percentage was used? If skipped, how many outliers were removed?

For all statistical results, it is necessary to include either df or N.

MRS

The authors state that good shims were obtained in the HC voxel for 18 of the 24 participants. It is necessary to state whether a specific line width threshold was used for exclusion of spectra, and if so, what threshold was used. It appears that the mean (s.e) of the linewidth is presented. Please present the mean (s.d.).

The authors state that 4 voxels were excluded for lipid contamination. Please clarify how many

were excluded from each voxel location.

The authors helpfully teach the reader that 2D-JPRESS offers some advantages over PRESS in regions of high inhomogeneity, like the HC. However, they fail to mention an apparent disadvantage of the 2D-JPRESS method when compared to the more commonly used MEGA_PRESS approach. Specifically, it appears from the cited JPRESS studies that the reliability of GABA/Cr measurements is considerably less with JPRESS than is typically reported for MEGA-PRESS. Given the HC target location, and the appearance of valid GABA measurements, this is not a criticism of the choice to use JPRESS. However, for readers familiar with MEGA-PRESS, the apparently lower reliability of the JPRESS approach should be mentioned among the limitations of the study.

The issue of the stability of HC GABA measurements is particularly relevant in the current study because of the interval between BOLD measures and the GABA measures with which they were correlated was relatively long (mean = 13 days). It is essential to also report the range of interval days. Are the authors aware of any data on the stability of MRS GABA measures in HC or other regions across intervals in the range occurring in this study? Even the mean value (13 days) is quite long, and this aspect of the design represents a limitation of the study that should be acknowledged.

If estimates of glutamate content and gray matter fraction are to be used as covariates, then the mean (s.d.) of these measurements must be reported. Since these are inherently noisy measurements, the reader will want to see some information about their distribution.

There is some confusion in the supplement about the duration of the MRS acquisitions. On line 145, it states "TR/TE=2400/31-229ms, DTE=2ms, 4 signal averages per TE step ... yielding a total acquisition time of 13 min 28 sec."

The math doesn't seem to add up. I get a total of 16 minutes for this acquisition. Similarly on line 153 it states "In addition, water unsuppressed 2D 1H MRS data were acquired from each voxel with 2 signal averages recorded for each TE step (acquisition time 3 min 28 sec)." However, I calculate a total of 8 minutes for acquisition, if there are the same number of TE steps. Please clarify.

Please clarify whether or not signal from the macromolecule multiplet at ~3.0 ppm is included in the GABA estimate from this method.

Minor point

In addition to the primary findings relating HC GABA to both HC BOLD response during suppression (negative correlation) and SIF (positive correlation), there is also a finding that HC GABA is negatively correlated with HC BOLD during retrieval. In fact, the correlation is stronger for retrieval than for suppression. The authors address this in a reasonable way. However, they may be missing an opportunity to clarify a parsimonious view of why both findings emerge. The authors correctly point out that bulk tissue GABA measurements in brain cannot distinguish between the various compartments in which the GABA is located. In fact, the great majority of GABA in HC and cortex is located in the cytoplasm of GABAergic interneurons. Cytoplasmic GABA serves, in part, as a reservoir both for the filling of synaptic vesicles with GABA and for extrasynaptic GABA release (as in tonic inhibition). Thus, some have argued that MRS GABA reflects the capacity for GABA-mediated effects during times of high demand (e.g. during tasks). It is quite possible that both retrieval and suppression evoke and depend on an increase in HC GABA-mediated effects. If so, then the BOLD response during both task components could be negatively associated with the bulk tissue GABA content in HC as measured by MRS. The association with bulk GABA does not distinguish between the specific GABA-mediated effects involved in the different tasks.

Reviewer #4 (Remarks to the Author):

Schmitz and colleagues reported a multimodal neuroimaging study in which they investigated how hippocampal GABA contributes to suppressing unwanted thoughts, with fMRI and 1H magnetic resonance spectroscopy (MRS). During fMRI scanning, 30 participants performed an adapted Think/No-Think (TNT) task and a Stop-signal (SS) task, which were interleaved in a mixed block/event-related design. 1H MRS data were obtained on a separate day to measure GABA concentrations in three regions of interest (ROIs), including the right hippocampus, the right dorsolateral prefrontal cortex (DLPFC) and the primary visual cortex. Three major results are reported: (1) fMRI data revealed that suppression led to reduced hippocampal activation and impaired memory for suppressed memories; (2) 1H MRS data revealed that greater hippocampal GABA concentrations predicted better mnemonic control in both retrieval and suppression conditions; (3) Higher hippocampal GABA specifically predicted stronger suppression-induced negative coupling between the DLPFC and the hippocampus. The authors concluded that GABAergic inhibition local to the hippocampus plays a critical role in mediating fronto-temporal inhibitory control pathway involved in the suppression of unwanted thoughts or memories.

Overall, there are several novel and significant strengths for this well-written manuscript, particularly the use of both fMRI and 1H MRS to address an important question of how hippocampal GABA contributes to suppressing unwanted memories in humans. It would be wise to publish this novel piece of work with no delay. The experimental design was very thoughtful and well controlled, involving a TNT task interleaved with a SS task. The authors have done a good job on including control regions in 1H MRS and conducting dynamic causal modeling analysis for fMRI data. The association of hippocampal GABA concentrations with hippocampal activation, functional coupling and dynamic causal interactions are very interesting. These findings will not only have important implications into understanding of neurobiological mechanisms underlying suppression of unwanted thoughts/memories, but also provide novel insights into understanding of intrusive symptoms of various psychiatric disorders. Despite of above novel and potentially important aspects, I do have several suggestions (detailed below) to improve the manuscript.

Major comments:

1. In the Introduction section, the authors emphasized several aspects of diminished lateral PFC engagement in cognitive control and hippocampal hyperactivity seen in a variety of psychiatric disorders. Although they attempted to build a link of local GABAergic inter-neuron network with hippocampal hyperactivity, it is still not that clear about the logic of how hippocampal local GABA actually modulates long-range PFC region(s) thought to drive top-down control over unwanted thoughts or memories. This point should be better framed to aid readers. For instance, the author may want to clarify this point by building up more thoughtful arguments about potential GABA neuromodulatory pathways acting on long-range PFC regions.

2. Another point related to above, the authors may want to point out how tonic hippocampal GABA network functioning may actually modulate their observed phasic hippocampal BOLD signals/activity and functional coupling with the DLPFC in their current fMRI study. This way may be helpful for readers to better understand the link of tonic high/low GABA concentrations with their observed effects on both behavioral and neuroimaging levels.

As they introduced that tonically disinhibiting GABAergic interneuron networks in the hippocampus has been linked to desynchronized hippocampal rhythms, reduced overall activity and impaired memory performance (line 41-42), one would thus expect to see an overall reduction pattern in hippocampal BOLD activity between high versus low hippocampal GABA groups. It would be great if the authors could look into their fMRI data about this point.

Did the author collect resting state fMRI data? It would be great to verify whether hippocampal

GABA is tonically related to task-free intrinsic hippocampal activity and intrinsic hippocampal-DLPFC connectivity at a resting rather than an active task state.

3. The central findings in this study are that hippocampal GABA levels were predictive of not only suppression-induced forgetting, but also BOLD hippocampal activity and connectivity as well as hippocampal-DLPFC dynamic causal interactions. Unfortunately, the authors did not report whether there was any potential difference in memory acquisition phase between high versus low hippocampal GABA groups. Based on above concern in Comment 2, one would expect that tonic hippocampal GABA concentrations might contribute to not only hippocampal-dependent memory processing not only during the suppression phase but also during the acquisition phase. This point is also somehow in line with their observed correlation with general memory performance regardless of Think/No-Think trials. It would be relevant to see any potential difference in memory performance between high versus low GABA groups during the training phase. They may simply compare training time and memory performance between during the TNT training phase between two groups.

4. In the training phase, participants were trained only to reach a learning criterion of at least 40% for the critical memories on the Think/No-Think task. What is the mean rate across participants? How much individual differences are there after this training procedure? In reality, however, there must be some participants reaching higher or lower than average. It is unclear this potential variance took into account for their analyses of fMRI data and 1H MRS data?

5. The authors have done a good job on analyzing hippocampal-DLPFC dynamic causal interactions and their links to local hippocampal GABA concentrations. This analytic approach looks only into hippocampal-DLPFC neural pathways while ignoring other potentially important neural pathways. As the authors have noted in the Introduction section, suppression of unwanted thoughts is most likely to carry out through polysynaptic pathways of the DLPFC to down-regulate hippocampal activity. The authors may want to point out this limitation in their manuscript.

6. The authors reported significant correlation of hippocampal GABA with suppression-induced forgetting, hippocampal activity and hippocampal-DLPFC functional coupling. It would be interesting to know whether there is any reliable moderate relationship among GABA, brain activity/functional coupling and memory performance. In other words, they may also want to consider GABA-brain-behavior moderation analysis (i.e., <https://github.com/canlab/MediationToolbox>) on the whole brain activity and hippocampal-based connectivity. This approach may provide some complimentary data to illustrate other possible modulatory pathways on the whole brain level.

7. For suppression-induced hippocampal BOLD activity, did the authors only look into No-Think trials regardless of subsequent memory status (i.e., later remembered or forgotten)? If memory status was considered, how did they differ while linking to hippocampal GABA concentrations? These data may be helpful to better understand the link of hippocampal GABA with suppression-induced forgetting and corresponding neural activity

8. In the Methods section, there appears no any description about 1H MRS data acquisition and analysis, fMRI data functional connectivity and dynamic causal modeling analyses. I would courage to include these parts in the Methods.

Minor comments:

9. More details are needed to aid readers about how regional GABA concentrations were computed for each ROI. For instance, it appears that three ROIs show quite different profiles for their frequency distribution of observed GABA concentrations in each voxel. How are the overall GABA

concentrations then computed each ROI?

10. In Figure legend S1: , I believe that "sagittal and axial slices" should be "sagittal and coronal slices".

11. On line 532: In the fMRI analysis section on line 589-690, the authors wrote as "Each model included within-session global scaling (default). Please clarify whether this is same as "global intensity normalization" implemented in SPM or not.

12. In the Supplemental Materials, it is unclear what the abbreviations of "SP and IP" on line 205 stand for.

Response to reviews: NCOMMS-16-24888

We thank the reviewers for the very high quality feedback we have received on this manuscript. We believe we have fully addressed the concerns raised, and this has led to a greatly improved paper.

Reviewer 1

Reviewer Comment 1.1: This paper uses functional MRI (fMRI) and Magnetic Resonance Spectroscopy (MRS) to study the brain mechanisms underlying control over unwanted thoughts. Many tests were made so as to infer that these mechanisms were specific both to the brain regions involved (e.g. hippocampus rather than primary motor/visual cortex) and what was being controlled (e.g. thoughts rather than actions). I find the paper to be impressive in the quality with which a broad range of techniques have been used and harnessed together to answer an important question (how does the brain suppress unwanted thoughts ?) - a question of relevance to many psychiatric disorders.

Author Response 1.1: We greatly appreciate reviewer 1's positive response.

Reviewer Comment 1.2: In what follows I will review each section of the results, covering the main findings, and focussing on methodology:

(1). Thought suppression engages a functionally specific hippocampal pathway

The GLM-based mass univariate analysis of the fMRI data (Fig 1) used a correction for multiple comparisons of cluster-level inferences using high cluster forming thresholds ($p < 0.001$) - see Fig 3. This is the correct use of the technique (c.f. recent controversy over the use of cluster level inferences in fMRI; Eklund et al. PNAS, 2015).

Author Response 1.2: We agree with Reviewer 1 that the cluster-forming threshold is correct for this type of analysis.

Reviewer Comment 1.3: Subjects performed a Think/No-Think task and a contrast of Think versus No-think identified left and right hippocampus, whereas a contrast Go versus Stop identified left and right M1. I'm not especially familiar with this literature - I expect the Go-versus Stop paradigm has been scanned many times using fMRI with similar results - is this correct ?

Author Response 1.3: Yes this is correct: The stop-signal task is very well established, both as a behavioural manipulation of motor inhibition and in the study of the neural basis of motor inhibition. See Logan et al., (1997) for a highly cited early description of the paradigm and its sensitivity to motor impulsivity, and e.g. Aron et al., (2014) for a review of the subsequent behavioural and neuroimaging research utilizing the stop-signal paradigm. See also Zandbelt & Vink (2010) for clear evidence that motor response stopping down-regulates activity in M1, as is observed here.

Author Action Taken 1.3: We have made this more explicit by modifying the text at line 110 to: "...participants also performed the stop-signal task, a well-established procedure for measuring the inhibition of motor actions (Aron et al., 2014; Logan et al., 1997)." Moreover, we now specifically cite a published example in which M1 is suppressed during motor response inhibition to illustrate the point (Zandbelt & Vink, 2010).

Reviewer Comment 1.4: Whereas, is this the first time Think/No-think has been scanned?

Author Response and Action Taken 1.4: The TNT memory inhibition paradigm has been evaluated in over 16 fMRI studies, (e.g., Anderson et al., 2004; Benoit and Anderson, 2012; Benoit et al., 2015; Butler and James, 2010; Depue et al., 2007; Gagnepain et al., 2014; Levy and Anderson, 2012; Gagnepain et al., in press). The current patterns are representative of typical findings. We have made this more explicit at line 80 by modifying the text to: “Previous work with the TNT paradigm establishes...”

Reviewer Comment 1.5: Additionally, DLPFC was activated during suppression of thoughts or actions.

(2) Hippocampal GABA predicts (i) reduced BOLD and (ii) successful thought suppression (i) Hippocampal GABA predicted hippocampal BOLD response during Think and No-Think tasks (more GABA, less BOLD) but not during Go or No-Go tasks (Actions). DLPFC and visual cortical GABA did not make these predictions. These inferences were made using correlations over subjects with bootstrapped confidence intervals.

Author Response 1.5: Yes this summary of our findings and methods is accurate.

Reviewer Comment 1.6: (ii) Hippocampal GABA predicted 'suppression induced forgetting' (impairment of later memory for suppressed items). Again inferences were made using correlations over subjects with bootstrapped confidence intervals. Looks fine.

Author Response 1.6: Yes this summary is correct.

Reviewer Comment 1.7: (3) Reduced hippocampal GABA compromises fronto-hippocampal network dynamics.

Think/No-think tasks modulated the (undirected) connectivity between hippocampus and DLPFC. Here DLPFC was found, and this inference made, using a whole brain search using the method known as Psycho-Physiological Interaction (PPI). Here the statistical threshold was not set using a whole brain correction, but a region of interest centred on the DLPFC (Fig 4a). This seems fine given its expected role (from work prior to this paper) in behavioural inhibition.

To test for directed changes in connectivity the authors then used Dynamic Causal Modelling (Fig 4d,e). Subjects were split into those with high versus low hippocampal GABA. For higher hippocampal GABA subjects, the best network model was one in which DLPFC provided input and "No-Think" task modulating connectivity between DLPFC and hippocampus. This was not the case for the low GABA group. This inference was made using the 'exceedence probability' measure - indicating which (of the tested) models was the most likely (frequently used) in the population from which the subjects were drawn. Again, the application of this methodology is sound.

Author Response 1.7: Yes, this summary is accurate. Thank you for the feedback on the appropriateness of our methods.

Reviewer Comment 1.8. SUMMARY

Overall, the findings suggest that GABAergic inhibition local to the hippocampus implements prefrontal control over intrusive thoughts. This finding is consistent with previous literature (e.g. reductions of the BOLD signal in hippocampus) but the additional use of MRS with fMRI nails this down to GABA. The data analyses have been conducted in an exemplary manner and

clearly support the findings.

Author Response 1.8: We greatly appreciate reviewer 1's positive response to the work and thank them for their efforts. We hope our responses to their comments are satisfactory.

Reviewer 2

Reviewer Comment 2.1: 30 young adults were recruited to the study. They performed a fMRI session, where they performed a Think/No-Think task, and the Stop Signal (SS) task, to investigate the relationship between GABAergic activity in the hippocampus and inhibitory control of unwanted thoughts. MRS data were acquired in a separate session from the right hippocampus, the right DLPFC and the visual cortex. The authors showed that hippocampal GABA was inversely related to BOLD signal suppression in the hippocampus in response to thought suppression. Appropriate controls were performed. This is an interesting study, which uses complex methodology, and was clearly performed with care and thought. The manuscript is well-written and guides the reader through the data well.

Author Response 2.1: We thank the reviewer for their nice remarks about the work.

Reviewer Comment 2.2: However, I am somewhat unconvinced by the specificity of the results, as discussed in detail below. In addition, MRS of the hippocampus is difficult and while the authors acknowledge this and have provided some data to reassure the reader of the quality of their spectra it is currently not possible to assess the data quality fully here, making it difficult to know how to interpret their results.

Author Response 2.2: We address these concerns below, where they are further elaborated.

Reviewer Comment 2.3: Major Points

1. The authors introduce the paper in terms of a number of psychiatric conditions and then test their hypotheses on healthy controls. It is not immediately clear to me that the mechanisms underlying the inhibition of intrusive thoughts in psychiatric disorders are the same as the mechanisms underlying the instructed inhibition of thoughts in healthy controls.

Author Response 2.3: The reviewer correctly notes that the current design did not study psychiatric populations, but rather healthy adults; as such, our conclusions do not directly relate to psychiatric populations. Indeed, our main goal was to understand the thought suppression mechanism as it normally operates as a way to highlight what might go wrong in some mental disorders.

Given the above, the main question is whether our experimental model of thought control is relevant to the control of intrusive thoughts in daily life. If the answer depended only on the current study, it might not be clear. Fortunately, there is much more data to go on, about which the reviewer may not be aware. Work with the current Think/No-Think paradigm (used in over 100 articles) supports its relevance as a model of the suppression of intrusive thoughts in clinical samples. Consider these examples.

1. **PTSD.** Using the current paradigm, Catarino et al. (2015, *Psychological Science*) found that people with PTSD show marked deficits in suppression-induced forgetting and that these deficits predict patients' intrusive symptoms as measured by standard clinical PTSD scales such as the Impact of Events Scale. Waldhauser et al., strongly replicated these findings in a MEG experiment with Somali refugees.

2. **Anxiety.** Using the current paradigm, Marzi et al. (2013, *Frontiers in Psychology*) demonstrated marked deficits in suppression-induced forgetting for people high in trait anxiety, especially for aversive

scenes. Analogously, Benoit, Davies, & Anderson (2016, *Proceedings of the National Academy of Sciences*) reported a similar relationship between suppression-induced forgetting of future worries and trait anxiety. In the latter instance, the same fronto-hippocampal network identified here was implicated.

3. **Depression.** Using this paradigm, Zhang, Liu, & Luo (2016, *Nature Scientific Reports*) found that depressed participants showed marked deficits in suppression-induced forgetting, relative to matched controls, and a diminished N2, an ERP component related to executive function. This finding echoes many similar suppression publications about depression reported by Hertel, Joorman, and Colleagues.

4. **Attention Deficit Disorder.** Using the current paradigm, Depue, Burgess, Willcut, & Banich (2010, *Neuropsychologia*) reported evidence for deficits in suppression-induced forgetting in ADHD, and associated deficits in engagement of lateral prefrontal cortex to suppress hippocampal activity.

5. **Rumination.** Using the current paradigm, Fawcett et al. (2015, *Journal of Behavioral Therapy and Experimental Psychiatry*) demonstrated, in a large sample (N = 100), a significant relationship between retrieval suppression and self-reports of rumination about unwanted thoughts in daily life, as measured by standard clinical scales used to measure the clinical symptom of rumination (the RRS).

6. **Thought Control Ability.** Using the current paradigm, Kuepper et al. (2014, *Journal of Experimental Psychology: General*) and Catarino et al. (2015, *Psychological Science*) found that people's self-reports of how well they control intrusive thoughts and memories in daily life, as measured by the *thought control ability questionnaire* (aka, the TCAQ scale), are well predicted by suppression-induced forgetting. The TCAQ is a standard clinical scale devised to measure individual differences in the ability to control intrusive thoughts that may be clinically relevant, and strongly predicts anxiety.

7. **Intrusive memories of Analogue Trauma.** Strebb et al. (2015) measured suppression-induced forgetting on the task used here, and also collected EEG. They then exposed participants to a traumatic video clip depicting an event that people find distressing. Over the next week, the participants kept diaries of intrusive thoughts about the film. After a week, they completed a clinical instrument (the Impact of Events Scale), which measures intrusion symptoms. Participants' success at suppressing retrieval during the task (i.e. suppression-induced forgetting and also the N2 ERP component) predicted the frequency and distress of trauma-film related intrusions, and people's PTSD score on clinical scales.

It is also worth noting the diversity of stimuli used. The foregoing designs have used simple word pairs, face-scene pairs, word-object pairs, word-line drawing pairs, and even, in some cases, people's own autobiographical memories. Both neutral and emotionally negative contents have been used as well. In general, these various materials *consistently identify a common pathway involving the right DLPFC and the down-regulation of hippocampal activity.*

So, empirical data exist that permit confidence in the generality of the processes we are measuring, and that suggest their clinical relevance.

Author Action Taken 2.3 The reviewer's comment gave us a clear appreciation that, in our effort to be economical in our presentation, we might have failed in our job at communicating the depth of support for our experimental model. If unaddressed, this would be a significant problem because some readers might have the same response. We therefore revised the manuscript to more fully articulate the evidence base supporting the relevance of this model (see e.g., Lines 80-93).

In addition, we further addressed the reviewer's concern by introducing a new paragraph in the final discussion that explicitly discusses the issue of generalization for readers to consider (see Lines 551-574). This paragraph acknowledges the limitation of using emotionally neutral word pairs while also making the case that the existing literature supports the potential relevance of this work. We thank the reviewer for highlighting this shortcoming of our exposition, which enabled us to strengthen our case.

Reviewer Comment 2.4: 2. Quantification of GABA from the hippocampus is difficult. I am reassured by the line-width reliability across the 3 voxels, but it would be useful to have some values for the fit for the GABA per se for all three voxels to determine the reliability of the measures.

Author Response 2.4: The spectral fitting methods used in this study enable the estimation of metabolite peak amplitudes, but it is not possible to directly estimate the uncertainty and reproducibility of these peaks without performing repeated measurements which we were not able to do given the already long acquisition times required. Alternatively, a common metric used to estimate the uncertainty on a metabolite measurement is the Cramer-Rao Lower Bound (CRLB) of variance. CRLB cut-off thresholds of 20-50% are widely used as metabolite rejection criteria in the literature. However, it should be noted that there are limitations to the interpretability of CRLB values, and a low CRLB does not guarantee an accurate or reproducible metabolite concentration and vice-versa (Kries and Boesch 2003, and Kries 2004). Therefore other quality control measures should be used alongside CRLB, including line-width and inspection of residual plots, both of which were considered in this study.

Quality control criteria using line-width were included in the original submission, and the mean line-width for the hippocampal ROIs were shown to be comparable to the other two ROIs in our original submission. Visual inspection of the raw 2D spectra and post-fitting residual plots revealed lipid contamination in four ROIs (three is the visual cortex and one in DLPFC). No unexplained features were identified in the residual plots for any of the hippocampal ROIs.

In the table below we provide the mean Cramér-Rao Lower Bound (CRLB) values (\pm standard error of the mean) for GABA for each of the three voxels.

	mean GABA CRLB	SEM
HIP	22.86	2.26
PFC	6.72	0.25
VIS	5.39	0.34

The CRLB values were higher in hippocampus due to the location of the voxel in an area of the brain with lower signal-to-noise ratio (SNR), a direct consequence of increased B0 susceptibility. In addition, to ensure specificity of the measurements, the hippocampal ROI was also smaller than the other two volumes (see Figures 2 and S1 Methods part IV.2.a), which also results in decreased SNR. However, we elected to include subjects that passed the quality assurance screening on our other metrics (N=18), which included the line-widths obtained from the higher-order shims, and visual inspection of the fit and residuals in the spectral plots produced for each voxel and subject.

The second reason is that the relationships reported in this manuscript were relatively unaffected when subjects were weighted according to their GABA CRLB values. Specifically, under the assumption that higher CRLB reflects lower quality data, we used weighted least squares regression to give each data point its proper amount of influence over the parameter estimates. Each subject was therefore precisely weighted by subtracting their CRLB from a constant value, ensuring all weights were positive values. Individuals with higher CRLB values therefore contributed proportionally smaller weights to the model. Below we show the standardized coefficients (betas) for the primary relationships demonstrated with hippocampal GABA, in linear regression models with and without the CRLB weighting.

	Unweighted	Weighted
HIP GABA / HIP BOLD	Beta	Beta
NT	-0.46	-0.41
T	-0.60	-0.54
HIP GABA / Behavior		
SIF	0.59	0.59
HIP GABA / PPI		
DLPFC	-0.61	-0.61

Of the observed significant relationships with hippocampal GABA, inference on only one relationship was affected by weighting with CRLB (NT GABA/HIP BOLD). The actual magnitude of this effect on the Beta value was, however, quite small. In general, the weighted least squares regression analyses demonstrate that our relationships did not change substantially when carefully adjusting the amount of influence of each datapoint over the parameter estimates according to CRLB.

Finally, a third reason not to exclude subjects according to CRLB comes from a recent review paper by Roland Kreis (Kreis, 2016), “The Trouble With Quality Filtering Based on Relative Cramér-Rao Lower Bounds”. In this paper, Kreis argues that removal of ¹H MRS data based on CRLB cut-points introduces selection biases into the data, and inflates Type II error. Kreis further concluded that “CRLB should not be used to eliminate bad MRS data – certainly not as sole criterion – because they may just reflect low levels of the measured quantity”. In this study, this point is illustrated by showing that rejection of subjects on the basis of a fixed CRLB threshold can lead to biases between a patient population and healthy controls, but we would argue the same argument to be true when comparing ROIs of different sizes and different brain areas affected by different artefacts.

Author Action Taken 2.4: We now describe the weighted least squares regression of CRLB, as well as a separate weighted least squares regression assessing the impact of line widths (Hz) of the hippocampal voxel, in the main text MRS methods, and report the results of both the unweighted and weighted regression models in the supplemental information.

Reviewer Comment 2.5: 3. I am not convinced that the hippocampal GABA measure is “specific”. The authors say that other “difficult non-memory tasks sometimes also reduce hippocampal activity” but this was not the case with the control task here. I do not think that this can be therefore claimed to be specific to the task in question. This issue should either be addressed in the discussion directly, and the interpretation amended accordingly, or a control experiment performed.

Author Response 2.5: The reviewer is correct to note that our specificity claim rests on a juxtaposition of our GABA/suppression finding to other difficult (non-suppression) tasks that also reduce hippocampal activity. Fortunately, we included the motor response inhibition task, in part, because it is exactly the sort of difficult task we had in mind. After running a one-sample t-test on the simple effect [Stop – Go], we can reassure the reviewer that our motor response inhibition task reliably reduced hippocampal activity, although this effect was smaller, relative to thought suppression. This is something that should have been included in the original submission, and the reviewer is correct to point it out.

Is this motor-stopping related reduction in hippocampal BOLD related to hippocampal GABA as well? Can a participant’s tendency for difficult tasks to reduce hippocampal activity explain our findings? We show that this motor-stopping task-induced reduction in BOLD is (a) uncorrelated with hippocampal GABA and (b) does not explain the significant relationship between GABA and BOLD during the retrieval suppression task. Indeed, we observed no change in the relationship between hippocampal

GABA and hippocampal activity during the retrieval suppression task, even when we controlled for reductions in activity in that structure during motor stopping in a partial correlation analysis. Moreover, the selectivity of this relationship of hippocampal GABA to memory function extends to the behavioural level as well. We observed that the relationship between hippocampal GABA and memory inhibition performance (SIF), if anything, is improved (see lines 367) when we controlled for motor stopping performance (SSRT) in a partial correlation analysis.

Author Action Taken 2.5: To address the reviewer’s comment, we now report the reduction in hippocampal activity during motor inhibition at lines 278-284, in the section exploring the functional specificity of relationships between hippocampal GABA and hippocampal BOLD response in the Think/No-Think and Stop-signal tasks. This finding adds force to the evidence that follows, establishing the specificity of our relationship of hippocampal GABA to BOLD response during the No-Think and Think conditions. We thank the reviewer for this suggestion, as it tightens our case.

Reviewer Comment 2.6: 4. The hippocampal GABA levels and the task data were acquired on 2 different days. What assurance can the authors give that either of these measures is sufficiently stable across time to make this an appropriate analysis approach. This is a particularly important question given the gender split – GABA is thought (though not definitively shown) to vary with the menstrual cycle. Was time of day controlled for? Stimulants? Sleep? Alcohol intake the previous night?

Author Response 2.6: Longitudinal ¹H MRS indices of GABA are reliable within cognitively healthy young adults at mean intervals of more than half a year, e.g. 229 ± 42 days (Near et al., 2014). Indeed, the observed magnitude of intra-subject variability was approximately the same as longitudinal ¹H MRS studies conducted at much shorter intervals, indicating that the majority of variance between timepoints arises from measurement error. These findings indicate ¹H MRS indices of GABA, in cognitively normal adults, reflect stable biological traits. Our decision to acquire fMRI and MRS in separate sessions also reflects a deliberate strategy to maximise data quality: In piloting the study, we found that the long acquisition times required to acquire MRS in multiple voxels (~1 hour) led to participant fatigue and discomfort and to a reduction in data quality (e.g. head motion) when combined with the fMRI acquisitions in a single session..

Nevertheless, we assessed whether the relationships reported in this manuscript were affected when subjects were weighted according to their inter-scan interval. Specifically, under the assumption that longer intervals reflect lower quality data, we used weighted least squares regression to give each data point its proper amount of influence over the parameter estimates. Each subject was precisely weighted by the number of days between the fMRI and MRS acquisition, by subtracting this interval from a constant to ensure positive values. Individuals with longer intervals therefore contributed proportionally smaller weights to the model. Below we show the standardized coefficients (betas) for the primary relationships demonstrated with hippocampal GABA, in linear regression models with and without Interval weighting.

	Unweighted	Weighted
HIP GABA / HIP BOLD	Beta	Beta
NT	-0.46	-0.50
T	-0.60	-0.63
HIP GABA / Behavior		
SIF	0.59	0.63
HIP GABA / PPI		
DLPFC	-0.61	-0.55

Of the observed significant relationships with hippocampal GABA, none were affected by weighting with Interval; in fact most were slightly improved. The weighted least squares regression analyses therefore demonstrate that our relationships did not change substantially when carefully adjusting the amount of influence of each datapoint over the parameter estimates according to Interval between the fMRI and MRS scans.

Author Action Taken 2.6: To address the reviewer's comments, we revised the manuscript in several ways. First, we now include in the main methods text additional information about our pre-scan screening form on lines 632, which instructed participants to refrain from alcohol or other psychoactive drugs in the 24-hour period prior to the scan. Participants were also screened for medical history indicators, such history with psychotropic medications, prior experience with mental health issues, or head injury. We did not, however, collect information from our female participants concerning the point they were at in their menstrual cycles. Finally, we also now cite the Near et al (2014) paper demonstrating the longitudinal reliability of GABA (see Lines 200-201) and describe the weighted least squares regression of Interval in the main text MRS methods and report the above table in the supplemental results.

Reviewer Comment 2.7: If I understand correctly the subjects were trained on the tasks prior to any of the imaging. Could it not therefore be the case that the levels of hippocampal GABA here reflect how well subjects were able to learn how to perform this task, rather than reflecting the ability to inhibit thoughts per se?

Author Response and Action Taken 2.7: In principle, yes, the reviewer could be right. The data suggest, however, that this is unlikely to be a concern. First, as reported in Table S1 in our supplement, there were no differences in memory performance on the word pairs *at the end of the training phase* (immediately before fMRI scans were acquired) across our Low and High GABA groups ($t = 0.63$, $p = 0.54$). Indeed, the correlation between GABA measurements and this index of initial word pair learning was not significant, $r = -0.095$, 95% CI: [-0.4994 0.4075]. More generally, as can be seen in Table S1 in the supplement, the two groups showed nearly identical performance on various measures from the stop-signal reaction time task, suggesting that on both memory and motor measures, the groups were comparable in their ability to learn and perform tasks. Given these observations, our data point to a specific relationship between suppression-induced forgetting and hippocampal GABA, not to the broad ability to learn the materials needed to do the task or to general features of participant performance.

Author Action Taken 2.7: The answer to the reviewer's question seems like it would be of interest to readers. To report the relevant findings, we have now inserted a new sentence at Lines 357-359, in which we report that there was no correlation between HC GABA and initial memory performance. In this sentence, we also steer readers more directly to Table S1 for further exploration of how GABA might relate to performance measures in general.

Reviewer Comment 2.8: There were relationships between hippocampal GABA and BOLD signal here and elsewhere, but the BOLD signal is complex and not well understood. Were any behavioural relationships demonstrated, either with GABA or BOLD? This would be very useful to understand the importance of this relationship.

Author Response 2.8: We agree with the reviewer that relationships to behaviour are helpful in understanding the data. We did observe a relationship between hippocampal BOLD and memory inhibition performance (suppression-induced forgetting; SIF). This is reported on lines 105-107 of the current manuscript. We also observed a relationship between hippocampal GABA and memory inhibition performance (SIF). This is reported on lines 350 and Table 1c. These functional relationships show that the ability to down-regulate a thought (as estimated from suppression-induced forgetting) is indeed linked to hippocampal down-regulation during suppression, and to hippocampal GABA, in line with the

hypothesis.

Reviewer Comment 2.9: The MRS methods are not given in the main body of the manuscript. Given the detail in which the behavioural and fMRI acquisitions are described this seems like an odd omission from the main text, particularly as the behavioural and fMRI acquisition and analysis is relatively standard while the MRS is certainly not.

Author Response 2.9: The reviewer raises a very good point. Too much of the spectroscopy methodology was relegated to the supplemental section in our original submission.

Author Action Taken 2.9: We have revised the manuscript to strike a better balance between the fMRI and spectroscopy methodology, particularly the post-processing steps. We have now added basic information about acquisition sequences used for both the fMRI and MRS data to the main body text Methods section Lines 766-774 and Lines 803-818). We have also now added information about the covariates used for each ROI (glutamate and grey matter) and descriptions of the various additional control analyses, e.g. weighted least squares regression (using CRLB, line width, and Interval). See lines 172-224 and supplemental results.

Reviewer Comment 2.10: Minor Points

1. I am not sure that the bins in the histograms in figure 2 are informative – smaller bins would give a more detailed distribution that would be more informative to the reader.

Author Response and Action Taken 2.10: We agree that the distributions in figure 2 may obfuscate the data somewhat. We have removed these plots, and replaced this with simple numerical descriptions of the distributions (means \pm standard deviation), which are more precise and easier to compare between regions (see lines 215-217).

Reviewer Comment 2.11: 2. Many readers will not be familiar with 2D MRS – it would be useful if the authors could expand the figure legend to figure 2 to explain the figures to the non-expert.

Author Response 2.11: We agree that more information should be given about the 2D plot, especially so that non-experts can understand the report better.

Author Action Taken 2.11: We have simplified the figure legend and clarified its components (lines 228-239). We further have attempted to improve Figure 2 itself to better visually capture how the metabolite concentrations are estimated from model fitting.

Reviewer 3

Reviewer Comment 3.1: The authors present intriguing evidence suggesting an association between hippocampal GABA content and suppression of memory retrieval in a paired associates task. The manuscript has many strengths, including the memory suppression paradigm, the fMRI approach, and the neural circuit models. In addition, the approach to measuring hippocampal GABA is commendable and uncommon thus far in the literature.

Author Response 3.1: We greatly appreciate reviewer 3's encouraging feedback.

Reviewer Comment 3.2: However, there are significant problems with the overall conceptual framework and with the approach to correlational analyses used in support of the principal aims, as well

as some concerns about the GABA measures. Until these issues are addressed, it is difficult to assess the overall impact of the work.

Author Response 3.2: We addressed these concerns below, where they are further elaborated.

Reviewer Comment 3.3: Conceptual Framework

The authors present potentially important evidence suggesting an association between hippocampal GABA content and suppression of memory retrieval in a paired associates task. However, the conceptual framework offered in the introduction and discussion focuses primarily on cognitive and clinical phenomena that lack a clear relationship to suppression of paired associate retrieval. Word retrieval is generalized here to represent “thinking,” “thought,” “intrusive thoughts” “intrusive symptomatology” and “awareness.” Relatedly, retrieval suppression is conceptualized as “thought suppression,” “suppression of intrusive memories,” and “control of awareness.” The Oxford dictionary’s first definition of thought is “An idea or opinion produced by thinking, or occurring suddenly in the mind.” It is true that paired associated word retrieval could be considered a simple subtype of thinking, but it is not generally considered a valid proxy for the complex, clinically relevant thought processes discussed at length in the paper. For example, the words “thought,” “think” or “thinking” are found 155 times in the manuscript, but only once in the list of references. The HC BOLD and GABA findings pertaining to retrieval and retrieval suppression are important on their own. However, these findings do not permit generalization to cognitively and phenomenologically distinct and more complex processes such as pathological worry, rumination, obsession and hallucination.

Author Response 3.3: We can understand why the reviewer might suspect that the cognitive and clinical phenomena of interest in this paper may lack a clear relationship to the suppression of paired associate retrieval, and why our findings might not permit generalization to complex processes like rumination, pathological worry, obsession and hallucination. Indeed, if we were in the reviewer’s position and this was the only study we were focusing on, we might also share this view. Data exists, however, that supports the relevance of this experimental model to the processes of interest here, and we apologize to the reviewer for not doing a better job at presenting this background in the paper.

Prior work with the current Think/No-Think paradigm (used in over 100 publications) supports its relevance as a model of the suppression of intrusive thoughts and memories in clinical samples. This evidence documents the generality of the phenomenon and its mechanisms and their relationship to clinical phenomena. First we summarise this evidence, and then discuss the analytic considerations.

Generalizability across Materials. Most early applications of the Think/No-Think paradigm (see, e.g. Anderson & Green, 2001) used word pairs of the sort used here. Like the reviewer, however, we also considered it important to establish the generality of the phenomenon and its cognitive and neural mechanisms. Over the last 16 years, suppression-induced forgetting (SIF) has been established with a broad variety of materials: Word-word pairs; word-scene pairs; object-scene pairs; word-line drawing pairs; face-scene pairs; face-word pairs, and word-object pairs. SIF has been found for both emotionally neutral and negative materials. Critically, SIF has been found with (a) autobiographical memories, and even (b) intrusive, *person-specific* worries about recurrently feared future events. In all cases, suppressing retrieval reduces the accessibility of the suppressed content, establishing a content-general phenomenon that appears relevant to complex constructs (e.g. worries).

Generalizability of the Neural Mechanism. At present, we are aware of 16 fMRI studies using the Think/No-Think procedure, and a similar number of ERP studies. These studies suggest a fronto-hippocampal inhibitory control pathway that supports retrieval suppression in a *materials general manner*. Frontally driven hippocampal modulation occurs for simple word pairs, face-scene pairs, person-specific *worries about the future* (e.g. Benoit, Davies, & Anderson, 2016, PNAS), and even complex,

upsetting and persistently intrusive autobiographical memories (Fawcett et al. in preparation). Effective connectivity evidence of this pathway (using Dynamic Causal Modelling) has been established for (a) word pairs (Benoit & Anderson, 2012; *Neuron*), (b) neutral face-scene pairs (Benoit et al. 2014; *JOCN*); (c) word-object pairs (Gagnepain, Henson, & Anderson, 2014; *PNAS*), and (d) aversive face-scene pairs (Gagnepain, Hulbert, & Anderson, in press, *Journal of Neuroscience*; for related findings without DCM, see also Depue, Curran, & Banich, 2007; *Science*; Liu et al., 2016; *Nature Communications*). Other regions (e.g. the amygdala) are also involved with emotional materials, and are actively down-regulated by the right DLPFC along with the hippocampus during retrieval suppression, as established by effective connectivity analysis (Gagnepain et al, in press, *Journal of Neuroscience*).

These data suggest that the pathway engaged to suppress retrieval of word pairs maps very well onto the pathway used to suppress upsetting images and memories—to the point that *the very same prefrontal cortex region* identified in a word pair study can be used as an *a priori ROI* for the analysis of an autobiographical memory study, recovering the full pattern of effective connectivity with the hippocampus. These data—which were unfortunately not highlighted in our initial submission--provide a good empirical grounding for optimism about the generalization of the current findings to a broader range of stimuli that the reviewer would consider to be more transparently relevant to clinical disorders.

Examples of Clinical Relevance: Of course, the generality of the phenomenon and neural mechanism need not imply its clinical relevance. It is reasonable and appropriate to consider whether the foregoing mechanism may be entirely irrelevant to how people control intrusive thoughts in daily life, and may in no way be related to clinical disorders.

The evidence does not, however, favor this conclusion. The current experimental model has been linked to most of the clinical phenomenon of interest in the current paper. Consider the examples below.

1. **PTSD.** Using an aversive object-scene version of the Think/No-Think paradigm, Catarino et al. (2015, *Psychological Science*) found that people with PTSD show marked deficits in SIF and that these deficits predicted patients' reported intrusions on clinical PTSD scales such as the Impact of Events Scale. Waldhauser et al., replicated these findings in an Magnetoencephalographic experiment with *Somali refugees* using emotionally neutral object-line drawing associations.

2. **Anxiety.** Using a face-scene version of the Think/No-Think paradigm, Marzi et al. (2013, *Frontiers in Psychology*) found marked deficits in SIF for people high in trait anxiety, especially for aversive scenes. Analogously, Benoit, Davies, & Anderson (2016, *Proceedings of the National Academy of Sciences*) reported a similar relationship between SIF for person-specific future worries and trait anxiety. In the latter instance, the same fronto-hippocampal network was implicated with an effective connectivity analysis (DCM) that was highly similar to the one used here.

3. **Depression.** Using a word-pair version of the Think/No-Think paradigm, Zhang, Liu, & Luo (2016, *Nature Scientific Reports*) found that depressed participants showed marked deficits in suppression-induced forgetting, relative to matched controls, and a diminished N2, an ERP component related to executive function. This finding echoes similar publications about retrieval suppression mechanisms in depression (using word pairs) reported by Hertel, Joorman, and colleagues in clinical journals.

4. **Attention Deficit Disorder.** Using a face-scene version of the paradigm, Depue, Burgess, Willcutt, & Banich (2010, *Neuropsychologia*) reported deficits in suppression-induced forgetting in ADHD, and associated deficits in engagement of lateral prefrontal cortex to suppress hippocampal activity.

5. **Rumination.** Using a neutral-word pair version of the Think/No-Think paradigm, Fawcett et al. (2015, *Journal of Behavioral Therapy and Experimental Psychiatry*) found, in a large sample (N = 100), a significant relationship between retrieval suppression and self-reports of rumination about unwanted

thoughts in daily life, as measured by standard scales used to measure rumination (the RRS).

6. **Thought Control Ability.** Using an object-scene version of Think/No-Think paradigm, Kuepper et al. (2014, *Journal of Experimental Psychology: General*) and Catarino et al. (2015, *Psychological Science*) found that people's self-reports of how well they control intrusive thoughts and memories in daily life, as measured by the *thought control ability questionnaire* (aka, the TCAQ scale), are well predicted by suppression-induced forgetting. The TCAQ is a standard clinical scale devised to measure individual differences in the ability to control *intrusive thoughts*, and strongly predicts anxiety.

7. **Intrusive memories of Analogue Trauma.** Streb et al. (2015) measured suppression-induced forgetting on the word-pair task used here, and also collected EEG. They then exposed participants to a traumatic video clip depicting an event that people find very distressing. Over the next week, the participants kept diaries of intrusive thoughts about the film. After a week, they completed a clinical instrument (the Impact of Events Scale), which measures intrusion symptoms. Participants' success at suppressing retrieval during the verbal paired associate task (i.e. suppression-induced forgetting and also the N2 ERP component) predicted the frequency and distress of trauma-film related intrusions, and people's PTSD score on clinical scales.

Retrieval Suppression as a Model of the Control of Intrusive Thought. It's very easy to understand the reviewer's skepticism about accepting forgetting on an episodic memory test for paired associates as a proxy for the ability to suppress thoughts in general. Clearly human thought is not just about episodic memory, and it is not, as a general rule, reducible to something as simple as associative retrieval, especially of simple word pairs. How then, could we feel justified in making the generalization that we make in the paper about the relevance of this work to clinically relevant intrusive thoughts?

It is important to consider the fact that the Think/No-Think task doesn't model *all* varieties of thought. Rather, it is intended to model processes involved in *perseverative thoughts* that spring to mind unbidden. As the reviewer notes, the Oxford English dictionary includes thoughts that "*occur suddenly in the mind*". When confined to this sense of "thought", we suggest that our method credibly indexes a process shared with the ability to control perseverative thoughts, as evident by the clear relationships of suppression effects to intrusive symptomatology just reviewed.

But why should this be true?

Addressing this is simpler than it might seem. The perseverative nature of involuntary, intrusive thoughts renders automatic memory retrieval a natural model of this situation: *If not from memory, from where would a repeated thought spring?* Recurring thoughts or ruminations clearly do have a memory component, and this likely involves hippocampal activity, an idea that converges with the role of this structure in mind wandering and the "default mode".

Moreover, many intrusive thoughts are involuntary images that clinical psychologists have flagged as critical in disorders including anxiety, depression, PTSD, schizophrenia, and bipolar disorder (see, e.g. Brewin, Gregory, Lipton, & Burgess, 2010 for a review of the evidence for this trans-diagnostic symptom). This form of recurring intrusive image has already been examined in the Think/No-think task and suppression of these experiences engages the same fronto-hippocampal network engaged during suppression of words (with additional suppression in visual cortex). Finally, worries about the future are instances of episodic future thinking, which Daniel Schacter and Donna Addis and colleagues have spent the last 5-10 years arguing involves activity in the hippocampus in service of scenario construction. Thus, the suppression of future worries can be modeled as the suppression of repeated intrusive images/scenarios generated initially during episodic prospection (please see Benoit, Davies, & Anderson, 2016, PNAS). Here too, the same fronto-hippocampal pathway identified in the current study has now been shown to be engaged when people suppress their imagination for future events.

The foregoing illustrates that the line between “intrusive memories” and “intrusive thoughts” is not altogether clear and that many if not most intrusive thoughts of clinical significance reflect involuntary retrievals. Indeed, this intimate linkage between thinking and retrieval is reflected in the name of our procedure, first introduced in 2001: *The Think/No-Think paradigm*. Thus, although suppressing automatic retrieval of simple pairs does differ in various respects from the particular phenomena of clinical interest, there are nevertheless core processes indexed by this task that are demonstrably relevant to these clinical symptoms, and that there are excellent analytical reasons for this.

The Upshot. The foregoing illustrates that we have reasonable empirical and theoretical grounding for adopting this simple task as a model of processes important to the clinical phenomena of main interest. There is a long-term historical effort behind the current study that lends credibility to its relevance. We hope these considerations clarify why we believe our generalization to be appropriate. We respect that the reviewer may still disagree, but we hope they will consider granting us the courtesy of allowing us to have a different view on this subject.

Author Action Taken 3.3: We have retained our conceptual framing in terms of intrusive thoughts because we believe that both empirical and theoretical considerations warrant this.

However, we accept responsibility for the fact that our initial submission invited the kind of reaction that the reviewer had, given that it did not represent the background evidence and considerations clearly enough. We therefore have elaborated on this background in the introduction, which can be found on (see e.g., Lines 80-93). Moreover, we now include a new paragraph in the discussion that raises the issue of generalizability for readers to consider (see Lines 551-574). We thank the reviewer for prompting us to do this, because we should not take for granted that readers will be aware of the literature behind this work and because it is appropriate for readers to reflect explicitly about whether generalization is warranted.

Reviewer Comment 3.4: A second type of conceptual error here is presenting the GABA differences as causal, when the findings are only correlational. The authors attribute a causal role to bulk measures of HC GABA (as measured by JPRESS) when they say “GABA enables,” “depends on GABA”, “GABA alters”, “low GABA compromises”, “GABA influences.” In fact, the study provides evidence for associations with bulk measures of HC GABA. Speculations about causal relationships should be minimized and clearly framed as speculation or hypotheses for future testing.

Author Response and Action Taken 3.4: The reviewer is correct. We agree that our treatment of the findings would be improved if we tried to maintain a clearer separation, throughout the text, between hypotheses about causality and statistical association. We have revised the results sections reporting the intermodal correlations with GABA, and, where applicable, toned down our language describing the findings accordingly. Elsewhere in the introduction and discussion, we have also toned down the implication of causality (e.g. see lines 10, 529, 622). We do, however, continue to include causal statements in our hypotheses and in *interpretative statements* based on the associations.

Reviewer Comment 3.5: The manuscript’s title incorporates both of these misleading conceptual frames, using the terms “GABA enables” and “unwanted thoughts.” In contrast, the study actually shows evidence that HC GABA is associated with volitional memory suppression.

Author Response 3.5: As noted above, we believe that the conceptual framing of our work in terms of thought suppression is justified, and is not in error. We considered revising the title of the manuscript to eliminate reference to the causal role of GABA in mediating the ability we are measuring. In the end, this

decision comes down to the function of the title—whether it is an empirical summary, or a conceptual interpretation that we wish to emphasise. We decided on the latter. Based on the evidence presented in the manuscript, we argue that hippocampal GABA may enable the suppression of intrusive thoughts. This is the idea we wish to preserve.

Author Action Taken 3.5. Although we have retained our title, we do agree that in the body of the manuscript, we should carefully separate statistical association from the causal interpretation we are attributing to it. In doing so, we will highlight the issue of causality for the reader, encouraging them to draw their own conclusions based on the data. We also included an explicit statement in the discussion frankly acknowledging that experimental manipulations of GABA are required to draw causal conclusions, unlike the correlational approach used here. Specifically:

(Lines 571-574)

“Ultimately, however, determining whether successful thought suppression relies on local hippocampal GABA requires a direct test of this generalization, together with experimental manipulations of GABA rather than the individual differences correlational approach used here.”

Reviewer Comment 3.6: Overstating and overgeneralizing the findings occurs in many places in the text. For example, line 10 states “In so doing, we isolate a fundamental mechanism enabling inhibitory control over thought: GABAergic inhibition of hippocampal activity.” “Isolate a fundamental mechanism” is much too strong a phrase, “enabling” is speculative, and “control over thought” is much too general to associate with HC GABA based on this study.

Author Response and Action Taken 3.6: We respect the reviewer’s goal of ensuring that our language is calibrated to the data. In response, we generally scrutinised the manuscript to see whether any of the language used was overstated or overgeneralized and we made modifications to tune our statements more precisely. Here are 3 examples to illustrate the sort of changes we made to act upon this request:

1. *Line 10.*

Previous “In so doing, we **isolate** a **fundamental** mechanism enabling inhibitory control over thought: GABAergic inhibition of hippocampal activity.”

Revised: “In so doing, we **provide evidence** for a mechanism...”

2. *Line 532.*

Previous: Our results point to GABAergic inhibition of hippocampal retrieval processes as a **key** mechanism **underlying the suppression of thought**.

Revised. Our results point to GABAergic inhibition of hippocampal retrieval processes as a **potential** mechanism that **enables such thoughts to be suppressed**.

3. *Line 625*

Previous: If so, the current work **establishes** a transdiagnostic framework that specifies one **important** computational reason why persistent intrusive thoughts emerge from hippocampal disinhibition.

Revised: If so, the current work **offers** a transdiagnostic framework that specifies one computational reason why persistent intrusive thoughts emerge from hippocampal disinhibition.

We generally made an effort to change things in the spirit of the reviewer’s recommendation, even if they didn’t specifically mention it. Nevertheless, it is possible that there remain some cases of language that the viewer might take a different view on. In fairness, we perhaps have a different view of our conceptual framework—a view that we believe has a solid evidence base. We hope that in the event that such cases arise, the reviewer will consider them honest differences of opinion and consider giving us latitude.

Reviewer Comment 3.7: Correlations

Line 202 states “Because the robust and partial correlation analyses yielded similar conclusions, we focus on the partial correlations for simplicity.”

The reader assumes that a study principally aiming to examine the association between HC BOLD and HC GABA would have an a priori statistical plan for testing this association. If so, which of these two approaches to correlation analysis was chosen a priori? All results should be reported using the a priori method, with secondary comments on the convergence or divergence of results found with an alternate method.

Author Response 3.7: We agree that an a priori statistical plan is essential. We assure the reviewer, however, that we conducted the robust and partial correlation analyses using a consistent a priori strategy throughout the entire manuscript. Indeed, the task design (using both motor and memory inhibition tasks) was selected to facilitate a particular a priori analysis approach. We acknowledge however that the results in the initial version were distributed widely, traversing multiple sections of the results in both the main body text and supplemental information, rendering this strategy somewhat hard to follow.

Author Action Taken 3.7: Based on the reviewer’s feedback, in the revision we have substantially revised the results sections describing the intermodal relationships with GABA in order to improve the clarity of our original a priori strategy. We have more fully and explicitly described our a priori strategy in the beginning of the intermodal section (see Lines 244-258). Crucially, we have also replaced Figure 3, which depicted only a subset of the intermodal relationships, with two comprehensive Tables (Tables 1 and 2, pages 18-19), which serve both as an organisational framework of our analysis strategy and also as a core repository for all of our primary individual differences analyses. Tables 1 and 2 are now consistently referred to in the results sections for each analysis step, as opposed to the diffuse reporting employed in the prior draft.

Reviewer Comment 3.8: As written, there is a confusing intermixing of robust and partial correlation approaches. For example, line 202 suggests that the results of the partial correlation analyses are presented in the main paper. However, line 214 indicates that CI are used for testing significance and cites the papers on robust correlations. This suggests that robust correlations are being reported for these comparisons. Again on line 353, the citations for robust correlations are given in a context where they appear to be reporting partial correlations.

Author Response and Action Taken 3.8: Author Action 3.7 addresses this concern. In brief, we more fully and explicitly described our a priori strategy, including uniform method of inference, in the beginning of the intermodal section (see Lines 244-258). We also include Tables 1 and 2, which succinctly describe when robust or partial correlation was used, how they were performed (covariates used, degrees of freedom, and the resulting relationship).

Reviewer Comment 3.9: In addition, there is a lack of consistency in how correlations are applied in the manuscript. Sometimes the authors provide direct comparisons between correlations, and sometimes they don’t. For example, the authors report that HC BOLD-GABA correlations are significant during memory task components and not significant during motor task components. They interpret this as a selective finding, but they omit direct comparison of the correlations across tasks. However, the authors include a direct comparison between correlations for a different contrast on lines 241-244. Sometimes they include the GO condition BOLD responses as covariates in relevant analyses (line 240-1) and sometimes they don’t (lines 224). The result is the appearance of selectively focusing on findings that support their model and not making sincere attempts to challenge or disprove the model. The relatively low power of the key contrasts (N=18) may have a role in this selective reporting.

Author Response 3.9: We agree with the reviewer that the mixture of partial correlation techniques and comparisons between correlated correlations (Meng's z) was inconsistent. We have removed the Meng's z comparisons entirely from this revision of the manuscript. To infer functional and anatomical specificity, we now instead employ a uniform partial correlation strategy throughout the manuscript. These are fully detailed in Tables 1 and 2. As described in Author Action 3.7, we have also substantially revised the Results section describing the intermodal relationships to better reflect the organization and consistency of reporting in the Tables (Lines 244-258). In all cases, we start by reporting the robust correlations, then the 'Control' partial correlations (controlling for sex, grey matter, and glutamate), then the partial correlations controlling for these covariates plus a covariate from the Stop signal task ('Functional Specificity') or from the DLPFC region of interest ('Anatomical Specificity'). See, e.g. lines .288-300, lines 301-323)

Reviewer Comment 3.10: What type of robust correlation was used? Was it bend, skipped, or some other? If bend, what percentage was used? If skipped, how many outliers were removed? For all statistical results, it is necessary to include either df or N.

Author Response and Action Taken 3.10: Author Action 3.7 addresses this concern. In brief, we more fully and explicitly described our a priori strategy, including the type of robust correlation conducted (skipped) and outlier detection method, in the beginning of the intermodal section (see Lines 244-258). Tables 1 and 2 describe, for each robust correlation analysis, the number of outliers removed and the degrees of freedom. The Tables also describe the degrees of freedom for all partial correlation analysis as well. We believe Tables 1 and 2 visually capture our a priori strategy, and comprehensively address the reviewer's concerns regarding the organization and clarity of the correlation results.

Reviewer Comment 3.11: MRS

The authors state that good shims were obtained in the HC voxel for 18 of the 24 participants. It is necessary to state whether a specific line width threshold was used for exclusion of spectra, and if so, what threshold was used. It appears that the mean (s.e) of the linewidth is presented. Please present the mean (s.d.). The authors state that 4 voxels were excluded for lipid contamination. Please clarify how many were excluded from each voxel location.

Author Response and Action Taken 3.11: We agree that this information would be useful to report. We excluded linewidths ± 3 SDs from the mean linewidth for a given voxel, in accordance with the recommendation of Waddell et al (2007). We have now added the SD to the mean and SEM values for the linewidths (see lines 188-192). For each voxel, an average 1D spectrum was produced by averaging the data across all repetitions and all TEs. These spectra were visually inspected by two independent raters (TWS and MMC) to determine whether they suffered from lipid contamination. In four cases, both raters identified a large unexpected peak on the right-hand side of the spectrum, with the left tail of that peak significantly displacing the baseline for the remaining peaks. All four voxels were subsequently excluded from further analysis. There were no cases of a spectrum being flagged for lipid contamination by one rater but not the other. We have broken down the excluded voxels according to their location (see lines 195).

Reviewer Comment 3.12: The authors helpfully teach the reader that 2D-JPRESS offers some advantages over PRESS in regions of high inhomogeneity, like the HC. However, they fail to mention an apparent disadvantage of the 2D-JPRESS method when compared to the more commonly used MEGA_PRESS approach. Specifically, it appears from the cited JPRESS studies that the reliability of GABA/Cr measurements is considerably less with JPRESS than is typically reported for MEGA-PRESS. Given the HC target location, and the appearance of valid GABA measurements, this is not a criticism of the choice to use JPRESS. However, for readers familiar with MEGA-PRESS, the apparently lower

reliability of the JPRESS approach should be mentioned among the limitations of the study.

Author Response and Action Taken 3.12: We thank the reviewer for this point. We agree that 2D JPRESS sequences remain somewhat more exotic in the literature compared to MEGA-PRESS sequences, rendering assessments of their reliability across studies somewhat less robust. We have added a point to this inherent limitation at the end of the MRS methods section (see lines 222-224).

Reviewer Comment 3.13: The issue of the stability of HC GABA measurements is particularly relevant in the current study because of the interval between BOLD measures and the GABA measures with which they were correlated was relatively long (median = 13 days). It is essential to also report the range of interval days. Are the authors aware of any data on the stability of MRS GABA measures in HC or other regions across intervals in the range occurring in this study? Even the median value (13 days) is quite long, and this aspect of the design represents a limitation of the study that should be acknowledged.

Author Response 3.13: Longitudinal ^1H MRS indices of GABA are reliable within cognitively healthy young adults at mean intervals of more than half a year, e.g. 229 ± 42 days (Near et al., 2014). Indeed, the observed magnitude of intra-subject variability in that study was approximately the same as longitudinal ^1H MRS studies conducted at much shorter intervals, indicating that the majority of variance between timepoints arises from measurement error. These findings indicate ^1H MRS indices of GABA, in cognitively normal adults, reflect stable biological traits. The interval between first and second visits in our study ranged from 1—111 days (mean \pm SD: 26 ± 34 days), which is considerably lower than Near et al., (2014). The decision to acquire fMRI and MRS in separate sessions also reflects a deliberate strategy to maximise data quality: In piloting the study, we found that the long acquisition times required to acquire MRS in multiple voxels (~1 hour) led to participant fatigue and discomfort and to a reduction in data quality (e.g. head motion) when combined with the fMRI acquisitions in a single session.

Nevertheless, we assessed whether the relationships reported in this manuscript were affected when subjects were weighted according to their inter-scan interval. Specifically, under the assumption that longer intervals reflect lower quality data, we used weighted least squares regression to give each data point its proper amount of influence over the parameter estimates. Each subject was precisely weighted by the number of days between the fMRI and MRS acquisition, by subtracting this interval from a constant to ensure positive values. Individuals with longer intervals therefore contributed proportionally smaller weights to the model. Below we show the standardized coefficients (betas) for the primary relationships demonstrated with hippocampal GABA, in linear regression models with and without the Interval weighting.

	Unweighted	Weighted
HIP GABA / HIP BOLD	Beta	Beta
NT	-0.46	-0.50
T	-0.60	-0.63
HIP GABA / Behavior		
SIF	0.59	0.63
HIP GABA / PPI		
DLPFC	-0.61	-0.55

Of the observed relationships with hippocampal GABA, none were affected by weighting with Interval; in fact most were slightly improved. The weighted least squares regression analyses therefore demonstrate that our relationships did not change substantially when carefully adjusting the amount of influence of each datapoint over the parameter estimates according to Interval between the fMRI and MRS scans.

Author Action Taken 3.13. In the main text MRS methods section, we now describe this as a limitation, report the range of days in the interval, its mean and SD (see Lines 199). We also now cite the Near et al (2014) paper demonstrating the longitudinal reliability of GABA (see Lines 200-201) and describe the weighted least squares regression assessing the impact of interval (see lines 201-213). The above table is reported in the supplemental results (V.1.a-c).

Reviewer Comment 3.14: If estimates of glutamate content and gray matter fraction are to be used as covariates, then the mean (s.d.) of these measurements must be reported. Since these are inherently noisy measurements, the reader will want to see some information about their distribution.

Author Response 3.14: Yes, we agree--this is a very good point. We now report the mean \pm SD for GABA, glutamate, and grey matter content, for each voxel in the main text MRS methods section (see lines 214-227).

Reviewer Comment 3.15: There is some confusion in the supplement about the duration of the MRS acquisitions. On line 145, it states “TR/TE=2400/31-229ms, DTE=2ms, 4 signal averages per TE step ... yielding a total acquisition time of 13 min 28 sec.” The math doesn’t seem to add up. I get a total of 16 minutes for this acquisition. Similarly on line 153 it states “In addition, water unsuppressed 2D 1H MRS data were acquired from each voxel with 2 signal averages recorded for each TE step (acquisition time 3 min 28 sec).” However, I calculate a total of 8 minutes for acquisition, if there are the same number of TE steps. Please clarify.

Author Response 3.15: This is indeed an error. Thank you for spotting this, the reported TR was incorrect, and should be 2000ms instead of 2400ms. In addition, for the water unsuppressed data only 1 average was acquired. We have corrected this text accordingly (see Lines 811-813).

Reviewer Comment 3.16: Please clarify whether or not signal from the macromolecule multiplet at ~3.0 ppm is included in the GABA estimate from this method.

Author Response 3.16: Yes, the basis functions in ProFit model all three of the GABA peaks, including the methylene group at 3.0 ppm. This is now clarified in the main text MRS methods section (see lines 180-182).

Reviewer Comment 3.17: Minor point

In addition to the primary findings relating HC GABA to both HC BOLD response during suppression (negative correlation) and SIF (positive correlation), there is also a finding that HC GABA is negatively correlated with HC BOLD during retrieval. In fact, the correlation is stronger for retrieval than for suppression. The authors address this in a reasonable way. However, they may be missing an opportunity to clarify a parsimonious view of why both findings emerge. The authors correctly point out that bulk tissue GABA measurements in brain cannot distinguish between the various compartments in which the GABA is located. In fact, the great majority of GABA in HC and cortex is located in the cytoplasm of GABAergic interneurons. Cytoplasmic GABA serves, in part, as a reservoir both for the filling of synaptic vesicles with GABA and for extrasynaptic GABA release (as in tonic inhibition). Thus, some have argued that MRS GABA reflects the capacity for GABA-mediated effects during times of high demand (e.g. during tasks). It is quite possible that both retrieval and suppression evoke and depend on an increase in HC GABA-mediated effects. If so, then the BOLD response during both task components could be negatively associated with the bulk tissue GABA content in HC as measured by MRS. The association with bulk GABA does not distinguish between the specific GABA-mediated effects involved in the different tasks.

Author Response 3.17: We agree that the association between GABA and BOLD in the Think and No-Think conditions cannot, by itself, say something specific about the GABA-mediated effects observed in the different tasks. We believe that these associations may arise either because (a) the GABA signal may be a proxy for the number of interneurons present in the volume, which may be associated with the ability to measure the functional effect of those interneurons on whatever processes they engage, or (b) the GABA signal may instead indicate the capacity to up-regulate GABA in times of high demand, as the reviewer notes, magnifying the impact of interneurons in whatever task they perform. In our thinking, both hypotheses emphasize the role of cytoplasmic GABA (given that synaptic and extracellular influences may make lesser contributions to the signal), but for different, and not mutually exclusive reasons. The moral of the story is that our view seems very consistent with the reviewer's. We do, however, believe that other analyses (e.g. the association of GABA to SIF and the relation of HC GABA to connectivity) do favour the idea that GABA may play differing roles in the retrieval and suppression tasks, consistent with our mechanistic hypothesis about interneuron disinhibition during suppression.

Author Action Taken 3.17: We have tried to capture the general idea suggested by the reviewer through revisions to paragraphs on Lines 327-371. Although we do not make specific reference to cytoplasmic GABA in this argument, we do emphasize the general idea that both retrieval and suppression may increase demands placed on GABAergic processes and that our relationships with BOLD cannot tell us about the nature of the underlying GABAergic process itself. We could have discussed in more detail the specific proposal that cytoplasmic GABA drives these effects, but we didn't think that this argument required that proposal, and opted for simpler exposition. We are very happy to change this description and make it more specific if the reviewer prefers it and thinks that is necessary.

Reviewer 4

Reviewer Comment 4.1: Schmitz and colleagues reported a multimodal neuroimaging study in which they investigated how hippocampal GABA contributes to suppressing unwanted thoughts, with fMRI and 1H magnetic resonance spectroscopy (MRS). During fMRI scanning, 30 participants performed an adapted Think/No-Think (TNT) task and a Stop-signal (SS) task, which were interleaved in a mixed block/event-related design. 1H MRS data were obtained on a separate day to measure GABA concentrations in three regions of interest (ROIs), including the right hippocampus, the right dorsolateral prefrontal cortex (DLPFC) and the primary visual cortex. Three major results are reported: (1) fMRI data revealed that suppression led to reduced hippocampal activation and impaired memory for suppressed memories; (2) 1H MRS data revealed that greater hippocampal GABA concentrations predicted better mnemonic control in both retrieval and suppression conditions; (3) Higher hippocampal GABA specifically predicted stronger suppression-induced negative coupling between the DLPFC and the hippocampus. The authors concluded that GABAergic inhibition local to the hippocampus plays a critical role in mediating fronto-temporal inhibitory control pathway involved in the suppression of unwanted thoughts or memories.

Overall, there are several novel and significant strengths for this well-written manuscript, particularly the use of both fMRI and 1H MRS to address an important question of how hippocampal GABA contributes to suppressing unwanted memories in humans. It would be wise to publish this novel piece of work with no delay. The experimental design was very thoughtful and well controlled, involving a TNT task interleaved with a SS task. The authors have done a good job on including control regions in 1H MRS and conducting dynamic causal modeling analysis for fMRI data. The association of hippocampal GABA concentrations with hippocampal activation, functional coupling and dynamic causal interactions are very interesting. These findings will not only have important implications into understanding of neurobiological mechanisms underlying suppression of unwanted thoughts/memories, but also provide novel insights into understanding of intrusive symptoms of various psychiatric disorders.

Author Response 4.1: We are quite grateful for reviewer 4's very positive feedback.

Reviewer Comment 4.2: Despite of above novel and potentially important aspects, I do have several suggestions (detailed below) to improve the manuscript.

Author Response 4.2: We thank the reviewer for their constructive suggestions, and we respond to them in turn, as they are addressed below.

Reviewer Comment 4.3: Major comments:

1. In the Introduction section, the authors emphasized several aspects of diminished lateral PFC engagement in cognitive control and hippocampal hyperactivity seen in a variety of psychiatric disorders. Although they attempted to build a link of local GABAergic inter-neuron network with hippocampal hyperactivity, it is still not that clear about the logic of how hippocampal local GABA actually modulates long-range PFC region(s) thought to drive top-down control over unwanted thoughts or memories. This point should be better framed to aid readers. For instance, the author may want to clarify this point by building up more thoughtful arguments about potential GABA neuromodulatory pathways acting on long-range PFC regions.

Author Response 4.3: We agree with the reviewer that it is pivotal that we make the proposed mechanism by which local GABAergic interneurons modulate long-range PFC connectivity as clear as possible for readers. We can see how, in trying to explain our rationale compactly, we might have sacrificed clarity for brevity.

To clarify, however, the reviewer's description above (i.e. that local hippocampal GABA affects activity in PFC regions) isn't quite the mechanism we are proposing. We do not think that local hippocampal GABA affects activity in the PFC (as suggested by the phrasing above); rather that the ability of the PFC to alter activity in the hippocampus is ultimately constrained by a final step—GABAergic interneurons in HC—through which top-down signals are implemented. Without these “boots on the ground” in the hippocampus, command signals will not have the desired effect. One need not speculate that PFC is affected at all in this scenario (even if it might be in disorders). The alteration in this final step, however, should affect connectivity between the PFC and HC.

Author Action Taken 4.3. As is clear from the reviewer's later point (4.5), they have a good understanding of the mechanism we have in mind. To communicate this mechanism more clearly, we revised the final paragraph (lines 50-69) in the introduction to take a little more time to make the point as lucidly as possible. We think that the mechanism is more clear now, with this elaboration. We hope that this addresses the reviewer's request.

Reviewer Comment 4.4: Another point related to above, the authors may want to point out how tonic hippocampal GABA network functioning may actually modulate their observed phasic hippocampal BOLD signals/activity and functional coupling with the DLPFC in their current fMRI study. This way may be helpful for readers to better understand the link of tonic high/low GABA concentrations with their observed effects on both behavioral and neuroimaging levels.

Author Response 4.4: The reviewer raises a good point. Even given greater clarity in describing the putative mechanism by which HC GABA alters PFC influence (as sought in request 4.3), it is a separate matter to also explain clearly how those mechanisms might translate into phasic changes in hippocampal BOLD and functional coupling. We have already sought to do this in the original manuscript, but we acknowledge that it could be more clear.

Author Action Taken 4.4: We have now tried to make our assumptions more explicit for readers. To achieve this, we added new text in the section where we first test relationships between hippocampal GABA and BOLD signal. In retrospect, we now see that we never explicitly stated what we believed this relationship would be, even though it was implicit in the argument. We now state this on Lines 262-270

“Prior work with non-human primates, combining fMRI with cortical electrophysiology, suggests that stimulus-induced negative BOLD responses in visual cortex arise, in part, due to increases in neuronal inhibition (Schmuel et al., 2006). Moreover, in humans the magnitude of task-induced negative BOLD responses in anterior cingulate have been linked with co-localized 1H MRS estimates of GABA concentration (Northoff et al., 2007; Walter et al, 2009). Together, these findings raise the possibility that negative BOLD responses in the hippocampus may also be linked with neuronal inhibition, and thus, co-localized 1H MRS estimates of GABA concentration. If so, our MRS measure of baseline GABA should predict reduced memory-driven BOLD responses arising during the Think/No-Think task.

This explicitly articulates our assumption that increasing GABA will be related to diminished task-related bold in a task-relevant region. In addition, we are now more explicit in breaking down the steps of the argument relating GABA to connectivity on lines 399-402, where we now state:

“If this fronto-hippocampal pathway provides afferent input that drives GABAergic processes during suppression, then how strongly DLPFC and hippocampus functionally integrate should depend on the availability of hippocampal GABA to implement retrieval stopping.”

One could attempt to speculate more precisely here about both mechanisms—e.g. by explicitly arguing for how increased tonic inhibition may alter local field potentials (and thus BOLD)—but we elected not to speculate at that level of specificity in the paper, but to keep the description of the putative relationship empirical.

Reviewer Comment 4.5: As they introduced that tonically disinhibiting GABAergic interneuron networks in the hippocampus has been linked to desynchronized hippocampal rhythms, reduced overall activity and impaired memory performance (line 41-42), one would thus expect to see an overall reduction pattern in hippocampal BOLD activity between high versus low hippocampal GABA groups. It would be great if the authors could look into their fMRI data about this point.

Author Response 4.5: We thank the reviewer for this keen observation. In general, the significant correlation we reported between hippocampal GABA and hippocampal BOLD during No-Think trials is consistent with this point, showing that people with higher GABA generally do show lower BOLD signal. However, when we do a median split into low and high GABA groups, the trend is in the expected direction (lower BOLD for higher GABA subjects), but it is not reliable. Overall these data do support the reviewer’s observation, however, and we feel that the continuous correlation is good evidence.

Reviewer Comment 4.6: Did the author collect resting state fMRI data? It would be great to verify whether hippocampal GABA is tonically related to task-free intrinsic hippocampal activity and intrinsic hippocampal-DLPFC connectivity at a resting rather than an active task state.

Author Response 4.6: These are terrific ideas. We wish we had collected resting state data, but we did not. Our focus was on comparing between two active task modalities of inhibitory control (memory versus motor), to draw inferences about functional specificity in the DLPFC—hippocampal pathway.

Author Action Taken 4.6: We definitely think that this idea is worth highlighting in the paper as a future direction. Thus, we added this to lines 620-624 of the paper, where we now say:

“This hypothesis suggests that estimates of hippocampal GABA should be related to hippocampal hyperactivity and to reduced resting state connectivity between the hippocampus and the prefrontal cortex; it may also partially account for the widely established difficulty in suppressing default mode network activity arising in a range of psychiatric disorders characterized by intrusive symptomatology.”

Reviewer Comment 4.7: The central findings in this study are that hippocampal GABA levels were predictive of not only suppression-induced forgetting, but also BOLD hippocampal activity and connectivity as well as hippocampal-DLPFC dynamic causal interactions. Unfortunately, the authors did not report whether there was any potential difference in memory acquisition phase between high versus low hippocampal GABA groups. Based on above concern in Comment 2, one would expect that tonic hippocampal GABA concentrations might contribute to not only hippocampal-dependent memory processing not only during the suppression phase but also during the acquisition phase. This point is also somehow in line with their observed correlation with general memory performance regardless of Think/No-Think trials. It would be relevant to see any potential difference in memory performance between high versus low GABA groups during the training phase. They may simply compare training time and memory performance between during the TNT training phase between two groups.

Author Response and Action Taken 4.7. We agree that a fuller picture of behavioural performance for each subgroup would greatly benefit the reader in interpreting the selectivity of the relationships of hippocampal GABA to memory suppression, versus more general learning capacity. We now report these data in Table S1 in our supplement. There were no differences in memory performance on the word pairs at the end of the training phase (immediately before fMRI scanning) across our Low and High GABA groups ($t = 0.63$, $p = 0.54$). Indeed, the correlation between GABA measurements and this index of initial word pair learning was not significant, $r = -0.095$, 95% CI: [-0.4994 0.4075]. We now report this on lines 354-356. More generally, as can be seen in Table S1 in the supplement, the two groups showed nearly identical performance on various measures from the stop-signal reaction time task, suggesting that on both memory and motor measures, the groups were comparable in their ability to learn and perform tasks. Given these observations, our data point to a specific relationship between suppression-induced forgetting and hippocampal GABA, not to the broad ability to learn the materials needed to do the task or to general features of participant performance.

Reviewer Comment 4.8: In the training phase, participants were trained only to reach a learning criterion of at least 40% for the critical memories on the Think/No-Think task. What is the mean rate across participants? How much individual differences are there after this training procedure? In reality, however, there must be some participants reaching higher or lower than average. It is unclear this potential variance took into account for their analyses of fMRI data and 1H MRS data?

Author Response 4.8: It should be emphasized that 40% was our minimum threshold of learning; most participants readily exceeded this. In fact, the mean performance after learning was 71.39 ± 3.48 s.e.m.% (median = 73%), which is typical of most TNT studies. Performance ranged from 42% to 97% for the whole group. As shown in supplemental Table S1, acquisition did not differ significantly between groups characterized by low and high GABA. Most importantly however, the correlation between hippocampal GABA measurements and this index of initial word pair learning was not significant, $r = -0.095$, 95% CI: [-0.4994 0.4075]. We now report this on lines 354-356.

Author Action Taken 4.8: Because we agree with the reviewer that readers may be interested in these points, we now report this relationship between hippocampal GABA and word pair learning (see Lines 357-360).

Reviewer Comment 4.9: The authors have done a good job on analyzing hippocampal-DLPFC dynamic causal interactions and their links to local hippocampal GABA concentrations. This analytic approach

looks only into hippocampal-DLPFC neural pathways while ignoring other potentially important neural pathways. As the authors have noted in the Introduction section, suppression of unwanted thoughts is most likely to carry out through polysynaptic pathways of the DLPFC to down-regulate hippocampal activity. The authors may want to point out this limitation in their manuscript.

Author Response 4.9: We acknowledge that our DCM is an oversimplification of the true underlying network. The choice of two regions was determined by the minimum set required to test our hypotheses of interactions between DLPFC and the hippocampus, extending previous neuroimaging models (Benoit et al., 2012). One could consider the inclusion of additional nodes. However, this is not necessary to determine the interactions between DLPFC and hippocampus, because DCM paths implicitly represent both monosynaptic and polysynaptic connections between nodes (Stephan et al., 2010). Nevertheless, we agree that future work on this issue may benefit from different and/or larger DCM networks, and that it is appropriate that we acknowledge this limitation in our discussion.

Author Action Taken 4.9: We added a sentence at the beginning of the final paragraph in the discussion (See Lines 606-608) in which we plainly acknowledge that we did not seek to identify the specific pathways through which the DLPFC modulates activity in either the hippocampus or the medial septal nucleus (we included mention of the latter for continuity from the prior paragraph). This addresses the reviewers' recommendation in a simple way.

Reviewer Comment 4.10: 6. The authors reported significant correlation of hippocampal GABA with suppression-induced forgetting, hippocampal activity and hippocampal-DLPFC functional coupling. It would be interesting to know whether there is any reliable moderate relationship among GABA, brain activity/functional coupling and memory performance. In other words, they may also want to consider GABA-brain-behavior moderation analysis (i.e., <https://github.com/canlab/MediationToolbox>) on the whole brain activity and hippocampal-based connectivity. This approach may provide some complimentary data to illustrate other possible modulatory pathways on the whole brain level.

Author Response 4.10: Thank you for this interesting suggestion. We did try moderation analysis on our data as the reviewer suggests, specifically a moderation model with: X=PPI estimates of fronto-hippocampal connectivity, Y=suppression-induced forgetting and M=hippocampal GABA. We did not find a significant moderation effect of GABA on the relationship between PPI and suppression-induced forgetting ($p>0.1$). However, we also tested a second moderation analysis with: X=hippocampal BOLD during No-Think trials, Y=suppression-induced forgetting and M=hippocampal GABA. Here we found a moderation effect of hippocampal GABA on the relationship between hippocampal BOLD response during No-Think and suppression-induced forgetting. However, the effect was small, $p=0.08$ two tailed.

Author Action Taken 4.10. Due to the small effect size and post hoc nature of these analyses, we have not included them in the current paper. However, we agree with the reviewer that conditional process models are a very interesting alternative for multimodal imaging studies.

Reviewer Comment 4.11: For suppression-induced hippocampal BOLD activity, did the authors only look into No-Think trials regardless of subsequent memory status (i.e., later remembered or forgotten)? If memory status was considered, how did they differ while linking to hippocampal GABA concentrations? These data may be helpful to better understand the link of hippocampal GABA with suppression-induced forgetting and corresponding neural activity.

Author Response 4.11: This is also a very interesting suggestion. Indeed, prior work by Depue and colleagues has focused profitably on the contrast between successfully forgotten No-Think items versus successfully remembered Think items. In this study, however, our analysis used all trials, irrespective of later memory outcome. One complicating factor in performing the analysis the reviewer suggests is that

we do not consistently have enough items across participants to bin trials according to subsequent forgetting. Given the overall high level of performance (in the 90% range for the Same Probe test), this means that there would be substantially fewer forgotten No-Think items than remembered No-Think items and not every participant would contribute to the analysis. This would likely render estimates of hippocampal activity quite noisy.

Reviewer Comment 4.12: In the Methods section, there appears no any description about ^1H MRS data acquisition and analysis, fMRI data functional connectivity and dynamic causal modeling analyses. I would courage to include these parts in the Methods.

Author Response 4.12: We agree with the reviewer that too much of the imaging analysis methodology was relegated to the supplemental section.

Author Action Taken 4.12: We have moved a substantial portion of the MRS analysis methods to the main body text, in accordance with this and other reviewer's request (see lines 172-227, 799-815). We have also moved some of the DCM analysis methods into the main body text (Lines 465-478).

Reviewer Comment 4.13: Minor comments:

More details are needed to aid readers about how regional GABA concentrations were computed for each ROI. For instance, it appears that three ROIs show quite different profiles for their frequency distribution of observed GABA concentrations in each voxel. How are the overall GABA concentrations then computed each ROI?

Author Response 4.13: The distributions are all multivariate normal. However, we agree that the distributions in figure 2 may obfuscate this point (see also reviewer 2's comment 2.10). We have removed these plots, and replaced this with simple numerical descriptions of the distributions (means \pm standard deviation), which are more precise and easier to compare between regions (see lines 215-217). We have also greatly expanded the main body MRS methods section to explain the process of estimating GABA concentrations from the model fitting (see lines 172-224). Finally we have improved Figure 2 to better visually capture how the model fitting is accomplished (see lines 228-239).

Reviewer Comment 4.14: 10. In Figure legend S1, I believe that "sagittal and axial slices" should be "sagittal and coronal slices".

Author Response 4.14: Thanks for this! Sorry for the confusion. We have corrected the error (see supplemental information, line 49).

Reviewer Comment 4.15: 11. On line 532: In the fMRI analysis section on line 589-690, the authors wrote as "Each model included within-session global scaling (default). Please clarify whether this is same as "global intensity normalization" implemented in SPM or not.

Author Response 4.15: The reviewer is correct; this is the same. This refers to the first level (single subject) design matrices. SPM's default option is 'None' (which we chose). From page 67 of the SPM12 manual <http://www.fil.ion.ucl.ac.uk/spm/doc/manual.pdf>:

"If you select "None" then SPM computes the grand mean value (formula), where N is the number of scans in that session. This is the fMRI signal averaged over all voxels within the brain and all time points within sessions. SPM then implements "Session-specific grand mean scaling" by multiplying each fMRI data point in session s by 100/gs."

Author Action Taken 4.15: We have now clarified this as “Session-specific grand mean scaling.” See line 786.

Reviewer Comment 4.16: 12. In the Supplemental Materials, it is unclear what the abbreviations of “SP and IP” on line 205 stand for.

Author Response and Action Taken 4.16: We have included explanations for these abbreviations, and the relevant citation should the reader require further explanation (see supplemental information, line 102).

REVIEWERS' COMMENTS:

Reviewer #2 (Remarks to the Author):

The authors have more than sufficiently addressed my concerns. Thank you for such a comprehensive and detailed response. The paper is now much improved - and is an important addition to the literature.

Reviewer #3 (Remarks to the Author):

The authors have responded with thorough, patient, and persuasive rebuttals. All of my significant concerns have been addressed in a fully satisfactory manner. The key conclusions are well-supported, and I think this will be a valuable contribution as currently written. I have two suggestions for the authors to consider if they wish.

1. I suggest including the mean and SD of the CRLB values for the hippocampal voxel in the supplement. I agree with Kries' point about the limited value of CRLB as a threshold for inclusion. However, reporting the observed values could be quite helpful for others who may attempt similar measurements in the hippocampus.

2. In the abstract and manuscript you refer to the "volitional control of awareness." I suggest there is an important distinction to be made between the control of "awareness" and the control of "the content of awareness." The former is not easily amenable to volitional control (short of consuming sleep-inducing substances). I think you mean to implicate control of the content of awareness. However, one can simply close one's eyes, and one has controlled the content of awareness. It seems to me that this study illuminates a mechanism involved in the volitional control over internally generated contents of awareness. As written, the idea that these findings relate to control over awareness itself seems to invite confusion.

Reviewer #4 (Remarks to the Author):

The authors have done a great job. They addressed most of my comments. They performed several additional analyses that confirm their initial results and improved the manuscript substantially. I have only some minor comments to improve the manuscript.

1. I still feel not that clear about how hippocampal GABA inhibition enables the ability of the DLPFC to exert long-range control over hippocampal retrieval processes. What I understood is following: It appears that the DLPFC initializes top-down signal acting on hippocampal retrieval processing and leading to the activation of hippocampal GABAergic interneuron networks which may selectively attenuate hippocampal functional coordination with ventral temporal regions where representation of a memory is typically coded.

If this makes sense, one would also expect relatively weaker hippocampal connectivity with temporal regions for memory suppression condition in individuals with high GABA concentrations.

2. With regarding to "inhibitory neural pathway" in memory suppression, the authors sometimes refer to "fronto-hippocampal inhibitory pathway" and sometimes refer to "fronto-temporal inhibitory pathway". These two terms may cause some confusion for the readers. Is there a better way to resolving this issue in the revised manuscript?

3. In lines 76-77, the authors state as following: "tonically disinhibiting GABAergic interneuron networks in the hippocampus desynchronizes hippocampal rhythms, reducing overall activity and impairing memory function". In other words, hippocampal GABA disinhibition leads to a reduced

overall activity in the hippocampus. How does this relate to “hippocampal hyperactivity” introduced in the first two paragraphs of the Introduction?

4. In lines 117-119 and 591-593, the authors put all citations at the end of sentence. I feel that it would be better to cite each reference corresponding to each specific type of stimuli. For instance, they may reorganize citations in the following way: “Suppression-induced forgetting occurs for a range of stimuli including words (Anderson et al., 2001, 2003; Benoit et al., 2012), visual objects (Gagnepain et al., 2014; Benoit et al., 2015), neutral and aversive scenes (Depue et al., 2007; Liu et al., 2016), autobiographical memories (Stephens et al., 2013) and even person-specific worries about feared future events (Benoit et al., 2016).” This would be helpful for readers to immediately catch up specific reference(s) pertaining to each type of stimuli.

5. As the authors point out in the manuscript, there is “a right lateralized fronto-hippocampal inhibitory pathway” engaged in memory suppression. Similar pattern was also reported by several studies on suppression of aversive memories (Depue et al. 2007 Science, Liu et al. 2016 Nature Communications). It appears that a very similar pathway may engage in memory suppression across a variety of stimuli or events. The authors may want to note this point and cite corresponding references in the revised manuscript.

6. In Figure legend S1, (D) (E) & (F) is not noted in the figure. The authors may want to go through the entire manuscript to avoid typos or error.

**Response to reviews:
NCOMMS-16-24888B**

We thank the reviewers for their helpful feedback on this manuscript. We believe we have fully addressed the remaining concerns raised.

Reviewer 2

Reviewer Comment 2.1: The authors have more than sufficiently addressed my concerns. Thank you for such a comprehensive and detailed response. The paper is now much improved - and is an important addition to the literature.

Author Response 2.1: We greatly appreciate reviewer 2's positive response.

Reviewer 3

Reviewer Comment 3.1: The authors have responded with thorough, patient, and persuasive rebuttals. All of my significant concerns have been addressed in a fully satisfactory manner. The key conclusions are well-supported, and I think this will be a valuable contribution as currently written. I have two suggestions for the authors to consider if they wish.

Author Response 3.1: We thank the reviewer for their kind remarks about the work.

Reviewer Comment 3.2: I suggest including the mean and SD of the CRLB values for the hippocampal voxel in the supplement. I agree with Kries' point about the limited value of CRLB as a threshold for inclusion. However, reporting the observed values could be quite helpful for others who may attempt similar measurements in the hippocampus.

Author Response and Action Taken 3.2: We have added the mean and SD of the CRLB values for the hippocampal voxel in the supplemental information.

Reviewer Comment 3.3: In the abstract and manuscript you refer to the "volitional control of awareness." I suggest there is an important distinction to be made between the control of "awareness" and the control of "the content of awareness." The former is not easily amenable to volitional control (short of consuming sleep-inducing substances). I think you mean to implicate control of the content of awareness. However, one can simply close one's eyes, and one has controlled the content of awareness. It seems to me that this study illuminates a mechanism involved in the volitional control over internally generated contents of awareness. As written, the idea that these findings relate to control over awareness itself seems to invite confusion.

Author Response and Action Taken 3.3: We agree with the reviewer's point. We have modified this clause to the "voluntary control over the contents of awareness" in the two instances in which it appears in the manuscript. See lines 34 and 527.

Reviewer 4

Reviewer Comment 4.1: The authors have done a great job. They addressed most of my comments. They performed several additional analyses that confirm their initial results and improved the manuscript substantially. I have only some minor comments to improve the manuscript.

Author Response 4.1: We greatly appreciate reviewer 4's encouraging feedback.

Reviewer Comment 4.2: I still feel not that clear about how hippocampal GABA inhibition enables the ability of the DLPFC to exert long-range control over hippocampal retrieval processes. What I understood is following: It appears that the DLPFC initializes top-down signal acting on hippocampal retrieval processing and leading to the activation of hippocampal GABAergic interneuron networks which may selectively attenuate hippocampal functional coordination with ventral temporal regions where representation of a memory is typically coded. If this makes sense, one would also expect relatively weaker hippocampal connectivity with temporal regions for memory suppression condition in individuals with high GABA concentrations.

Author Response and Action Taken 4.2: We did not test interactions between the hippocampus and surrounding temporal cortical regions. Although this is a very interesting question in and of itself, it is beyond the scope of the current paper. We only test and draw inferences about the a priori ROIs defined by our MRS acquisitions within DLPFC and hippocampus. We have modified the text to clarify that we did not test temporal cortical areas outside of the hippocampus. Please see the next comment.

Reviewer Comment 4.3: With regarding to “inhibitory neural pathway” in memory suppression, the authors sometimes refer to “fronto-hippocampal inhibitory pathway” and sometimes refer to “fronto-temporal inhibitory pathway”. These two terms may cause some confusion for the readers. Is there a better way to resolving this issue in the revised manuscript?

Author Response and Action Taken 4.3: We intended the terms “fronto-temporal” and “fronto-hippocampal” to refer to the same pathway. However, we now see that this was confusing. We have corrected this inconsistency and now only make reference to the “fronto-hippocampal inhibitory pathway.” which accurately reflects the regions of interest we tested with our ROI, connectivity, and MRS measures.

Reviewer Comment 4.4: In lines 76-77, the authors state as following: “tonically disinhibiting GABAergic interneuron networks in the hippocampus desynchronizes hippocampal rhythms, reducing overall activity and impairing memory function”. In other words, hippocampal GABA disinhibition leads to a reduced overall activity in the hippocampus. How does this relate to “hippocampal hyperactivity” introduced in the first two paragraphs of the Introduction?

Author Response and Action Taken 4.4: Hippocampal hyperactivity refers to the pathological state hypothesized to result from reduced tonic inhibition. In the above, we are describing the normal function of this network in the context of successful inhibition of thought, hypothesized to result from increased tonic inhibition. We have added text clarifying the relationship between these ideas. Specifically, on line 57 we added the sentences below to illustrate how, in healthy individuals, GABA helps to suppress thoughts, but when GABA is deficient, thoughts are hard to suppress, and hippocampal activity is higher.

Line 57: *“This same GABA deficit should also cause elevated hippocampal activity (hippocampal hyperactivity), explaining the recurring association between this feature and intrusive symptomatology.”*

Reviewer Comment 4.5: In lines 117-119 and 591-593, the authors put all citations at the end of sentence. I feel that it would be better to cite each reference corresponding to each specific type of stimuli. For instance, they may reorganize citations in the following way: “Suppression-induced forgetting occurs for a range of stimuli including words (Anderson et al., 2001, 2003; Benoit et al., 2012), visual objects (Gagnepain et al., 2014; Benoit et al., 2015), neutral and aversive scenes (Depue et al., 2007; Liu et al., 2016), autobiographical memories (Stephens et al., 2013) and even person-specific worries about feared future events (Benoit et al., 2016).” This would be helpful for readers to immediately

catch up specific reference(s) pertaining to each type of stimuli.

Author Response and Action Taken 4.5: We have modified the citation embedding for this sentence in accordance with the reviewer's proposal. See Lines 88-90 and 516-519.

Reviewer Comment 4.6: As the authors point out in the manuscript, there is “a right lateralized fronto-hippocampal inhibitory pathway” engaged in memory suppression. Similar pattern was also reported by several studies on suppression of aversive memories (Depue et al. 2007 Science, Liu et al. 2016 Nature Communications). It appears that a very similar pathway may engage in memory suppression across a variety of stimuli or events. The authors may want to note this point and cite corresponding references in the revised manuscript.

Author Response and Action Taken 4.6: We have added these citations to this sentence. See Line 518.

Reviewer Comment 4.7: In Figure legend S1, (D) (E) & (F) is not noted in the figure. The authors may want to go through the entire manuscript to avoid typos or error.

Author Response and Action Taken 4.7: We greatly appreciate the reviewer's careful proof-reading. We have corrected the figure legend for supplementary Figure S1, and proof-read the manuscript again in its entirety.